# Highly selective covalent organic functionalization of epitaxial graphene

Rebeca A. Bueno[1], José I. Martínez[1], Roberto F. Luccas[1,2], Nerea Ruiz del Árbol[1], Carmen Munuera[1], Irene Palacio[1], Francisco J. Palomares[1], Koen Lauwaet[1], Sangeeta Thakur[3], Jacek M. Baranowski[4], Wlodek Strupinski[4], María F. López[1], Federico Mompean[1], Mar García-Hernández[1] & José A. Martín-Gago[1]

Graphene functionalization with organics is expected to be an important step for the development of graphene-based materials with tailored electronic properties. However, its high chemical inertness makes difficult a controlled and selective covalent functionalization, and most of the works performed up to the date report electrostatic molecular adsorption or unruly functionalization. We show hereafter a mechanism for promoting highly specific covalent bonding of any amino-terminated molecule and a description of the operating processes. We show, by different experimental techniques and theoretical methods, that the excess of charge at carbon dangling-bonds formed on single-atomic vacancies at the graphene surface induces enhanced reactivity towards a selective oxidation of the amino group and subsequent integration of the nitrogen within the graphene network. Remarkably, functionalized surfaces retain the electronic properties of pristine graphene. This study opens the door for development of graphene-based interfaces, as nano-bio-hybrid composites, fabrication of dielectrics, plasmonics or spintronics.

[1] Materials Science Factory, Instituto de Ciencia de Materiales de Madrid-CSIC, C/Sor Juana Inés de la Cruz 3, Madrid 28049, Spain. [2] Instituto de Física Rosario-CONICET-UNR, Bv. 27 de Febrero 210bis, Rosario S2000EZP, Argentina. [3] Sincrotrone Trieste, strada Statale 14 - km 163, Basovizza 5 34149, Italy. [4] Institute of Electronic Materials Technology, Wolczynska 133, 01-919 Warsaw, Poland. Correspondence and requests for materials should be addressed to J.A.M.-G. (email: gago@icmm.csic.es).

Graphene, one of the most relevant materials of the decade, has to strengthen several drawbacks before it may step from fundamental physics to technology. Particularly, the absence of an electronic band-gap and its extreme chemical inertness significantly compromise their use as an active element in electronic devices. Development of highly precise and selective chemical functionalization strategies is understood as a break-through to circumvent this issue. A ruled covalent functionaliza-tion will provide added value to pristine graphene to develop applications in domains as important as band-gap engineering, creation of magnetism in graphene for applications in spintronics, fabrication of dielectrics, formation of nano-bio hybrid compo-sites for improved biosensors or tailoring optical-plasmonic properties. To this aim, wet chemistry protocols (among them, the use of diazonium salt, click chemistry or surface reactions) have been settled[1–13]. Unfortunately, many of the reported covalent modification methodologies require relatively harsh conditions, as high temperatures or long reaction times that are usually incompatible with some functional groups and may lead to functionalized surfaces with large number of defects that shrink the remarkable electronic properties of graphene. Moreover, only few reactions schemes are available and most of them are promoting direct C–C bond forming reactions with aryl addition[8,9]. Other kind of works use ultra high vacuum environments and are usually limited to small molecules, as hydrogen, oxygen or more recently cyanomethyl radicals[14–16]. This kind of works is usually performed for epitaxial graphene grown on metal surfaces and take advantage of the surface patterning obtained by lattice-mismatched graphene on metallic substrates. Nowadays, lack of selectivity is one of the major drawbacks on the covalent modification protocols[16].

For our highly selective covalent organic functionalization we used high quality grown epitaxial graphene on 4H-SiC(0001) by used chemical vapour deposition at $1,600\,^{\circ}\mathrm{C}$ under an argon (Ar) laminar flow in a hot-wall Aixtron VP508 reactor[17]. The single-layer graphene (SLG) produced by this method on the buffer layer presents large crystalline atomic terraces and it is much less sensitive to SiC surface defects, resulting in higher electron mobility than those grown by Si sublimation process. Moreover, intercalation of hydrogen under the buffer layer of 4H-SiC(0001) is known to convert the buffer into graphene, the so called Quasi Free Standing Monolayer Graphene (QFSMG)[18,19]. The QFSMG samples used for this study present Hall mobility values above $8,000\,\mathrm{cm^2\,V^{-1}\,s^{-1}}$. The present work has been performed using both, SLG and QFSMG. Both surfaces exhibit similar results, and therefore, we will refer to them indistinctly.

To validate our mechanism, we have chosen p-aminophenol (p-AP) as a model molecule as it incorporates an aromatic ring and two different ending groups. The only requirement for our methodology is to use a molecule terminated in an amino group, as the mechanism we will describe hereafter is selective to this particular group. The terminal group is the one that has to be chosen paying attention to the possible applications. In our case the phenol-linked molecule can be considered as a chemical spacer or linker for anchoring more complex molecular nanoarchitectures in the search of designing functional structures[2,3,6,10,20,21]. Moreover, several variations of this molecule are nowadays commercially available or possible to synthetize by several laboratory routes. We have experimentally confirmed that our protocol perfectly works also with aminothiophenol molecules and p-phenylenediamine molecules, confirming the general validity of this new route.

In this work we show an unprecedented chemical mechanism, described at an atomic level, by which the excess of charge located in unsaturated dangling bonds of C atoms around single-atomic vacancies (SAV) induces a selective oxidation reaction of any molecule containing an amino group, in which the N gets incorporated into de graphene lattice followed by a redistribution of the charge of the molecule in the neighbourhood of the vacancy. This mechanism can be used as a rational functionaliza-tion strategy to covalently couple amino-terminated organic molecules, as p-AP, to the surface of graphene epitaxially grown on SiC without disrupting its outstanding electrical properties over mm distances. Our method is based on making use of the localized electrons appearing around atomic vacancies. The physics of induced point defects on graphene have emerged in the last years as a source of new controlled properties, giving rise to magnetism[22], chemical properties[23], catalysis[24] or electronic doping[25]. Here we take advantage of the accumulation of charge at the dangling bond positions to promote new chemical reactions, as the selective binding of amino-terminated molecule. The results we show hereafter pave the way for the use of graphene as part of electronic circuits and for the formation of effective sensors and biosensors[26] or advanced field-effect transistors[27,28].

## Results

**Functionalization strategy and mechanism.** Figure 1a shows a scanning tunnelling microscope (STM) image of the clean surface of a SLG taken in an ultra high vacuum (UHV) chamber at room temperature (RT). Underneath the atomic-resolved graphene hexagons, bumps corresponding to the $(6\sqrt{3}\times6\sqrt{3})\mathrm{R}30^{\circ}$ reconstruction or buffer layer can be observed. STM images of QFSMG do not show the reconstruction in good agreement with previously published results[29]. Our functionalization protocol starts forming SAV in the graphene network by low-energy $\mathrm{Ar}^{+}$ ion irradiation. These vacancies have been previously observed and isolated by transmission electron microscopy (TEM) and STM, and their formation protocol is well established[29–35]. SAVs on the surface are easily identified by STM even at RT, because they appear as a threefold irradiating protrusions due to an increase in the local density of states at the Fermi energy, which is spatially localized on the dangling bonds of the neighbouring atoms[35,36]. Some of them can be observed in Fig. 1b, where the characteristics irradiating threefold channel are observed. These vacancies are stable and we have not observed then to diffuse or change upon annealing.

We have performed First-principles density functional theory (DFT)-based calculations on a fully relaxed model of 4H-SiC(0001) SLG/$(6\sqrt{3}\times6\sqrt{3})\mathrm{R}30^{\circ}$ with a unit cell with $13\times13$ SLG on top[37]. However, to make calculations more manageable we have removed the SiC substrate and left the buffer layer saturating the broken bonds with hydrogen atoms (Fig. 1e and Supplementary Note 1).

Regarding the SAV structural morphology, it has been recently shown that two distinct configurations of a SAV can be found in graphene[32,33]: a symmetric monovacancy (s-SAV), with three under-coordinated C atoms that each bonds to only two neighbours; and a Jahn–Teller reconstructed SAV (r-SAV), arising from the combination of the saturation of two initial dangling bond states in the reconstruction and an apparent out-of-plane displacement of the third under-coordinated C radical, very close in energy with respect to the s-SAV case. Aberration-corrected TEM (AC-TEM) at $80\,\mathrm{kV}$(ref. 32) and DFT calculations[33] suggest that Jahn–Teller r-SAV reconstruction seems to arise in graphene environments with a graphene strain above a 2%. On the other hand, AC-TEM experiments[32] report a minimum distance between dangling C atoms of $2.2–2.4\,\text{Å}$ and around $1.9\,\text{Å}$ for the s-SAV and r-SAV cases, respectively. In the present study, our first-principles calculations of the SAV shows a quasi-symmetric configuration s-SAV (Fig. 1d,e), with a

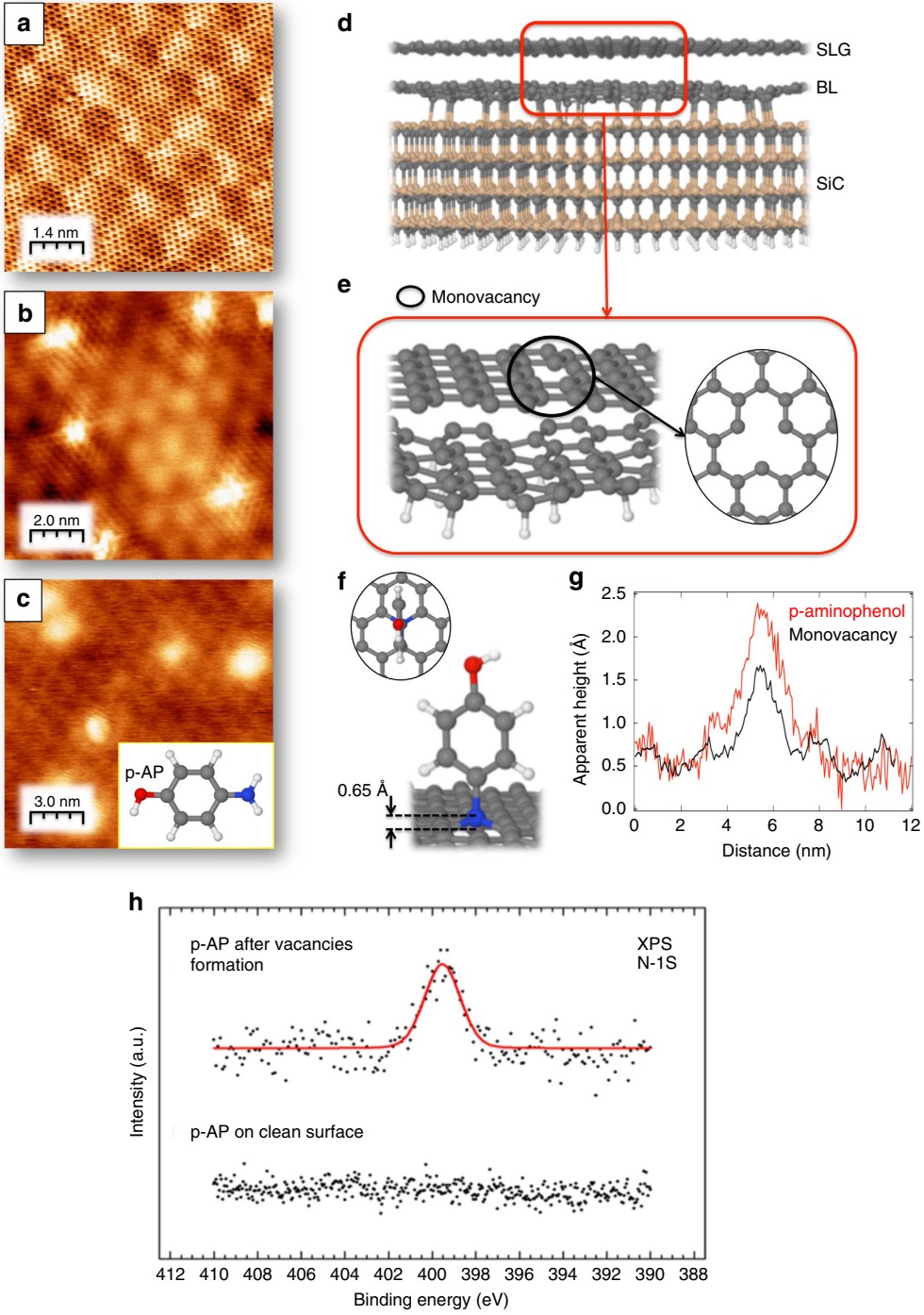

**Figure 1 | Covalent link of p-AP molecules on the surface via single-vacancy formation.** (**a**) Atomic-resolved STM image of the $(6\sqrt{3} \times 6\sqrt{3})R30°$ reconstruction of the SLG epitaxially grown on SiC(0001); Tunnel current and bias are 0.36 nA, −510.9 mV, respectively. (**b**) STM image of single atom vacancies formed upon Ar$^+$ irradiation on the surface; Tunnel current and bias are 2 nA, −289 mV, respectively. (**c**) STM image obtained after dosing the vacancies with 4L of p-AP; Tunnel current and bias are 0.09 nA, −1,240 mV, respectively. Images a and b were recorded with a H-terminated tip, whereas c, with a standard W tip. (**d**) pictorial representation of a fully relaxed model of 4H-SiC(0001) SLG/$(6\sqrt{3} \times 6\sqrt{3})R30°$ unit cell[37]; (**e**) top and side views of the optimized model for a single-vacancy created within a SLG on a graphene buffer layer. In this model we have saturated with hydrogen atoms the anchoring points between the graphene buffer layer and the SiC(0001) with respect to the model shown in **d**; (**f**) top and side views of the optimized geometry of a doubly-dehydrogenated p-AP molecule with the nitrogen atom integrated in the SLG lattice within a monovacancy. Average perpendicular-to-the-surface distance between the nitrogen atom and the SLG is also shown. Low energy Ar-sputtering was used to create a controlled density of atomic defects on graphene lattice; (**g**) pypical STM Profiles on images b and c, along a single-atom vacancy and a p-AP anchored molecule, respectively; (**h**) N 1s core level spectra after dosing p-AP on a SLG surface without and with atomic vacancies. The *N* 1s binding energy corresponds to that of a C atom substitutional in the graphene-network.

minimum distance between dangling C atoms of 2.27 Å (to be compared with the distance of 2.46 Å between two alternating C atoms in pristine graphene), in good agreement with the AC-TEM experimental evidence[32]. This observation is also reinforced by the fact that the strain experimented by the graphene on our buffer layer model is about 2%, which favours the formation of a s-SAV configuration, also agreeing with previous DFT calculations[33].

We observe that the apparent height of the SAV in STM images is about $1 \pm 0.2$ Å and its width about $1 \pm 0.2$ nm. These values are unaltered for the case of the QFSMG surface and are in good agreement with previously published values[31].

Once SAVs are created they appear stable and spread all over the graphene surface. The next step is dosing about four Langmuirs of p-AP. STM images recorded after this process show that the surface topography has importantly changed. All the localized threefold radiating 1 Å high bumps become larger and higher, losing their starring appearance. The average apparent height is now $1.3 \pm 0.6$ Å (Fig. 1c–g). Upon functionalization the width also increases and typically become about $1.4 \pm 0.6$ nm, value sensitive larger than that of the SAV. The larger data dispersion is related to the slightly variation of the apparent height with the bias potential, which in a first approximation, and in the range of accessible bias voltages, is independent of weather the tip is functionalized or not. All these values are statistically analysed in the Supplementary Fig. 1 and the Supplementary Note 2. The large widths depicted by STM images will be analysed in the next section in terms of electronic modification upon absorption. The surface morphology depicted in Fig. 1c is very similar to that reported for functionalized graphene on SiC (ref. 38).

The X-ray photoelectron spectroscopy (XPS) spectra recorded on the surface after dosing are indicative of the existence of N and carbonaceous species on the surface, suggesting that the bumps depicted in STM images corresponds to p-AP molecules anchored to the SAVs with an unique adsorption site for N atoms (Fig. 1h). The binding energy of the N 1s core-level peak derived from Fig. 1h is 399.5 eV. Pure N in a graphitic network has been reported to appear at 400.5 eV (ref. 39), however, in our case, although the N is occupying a substitutional place, it remains bonded to the molecule in a kind of $sp^3$ configuration, and therefore a lower binding energy value can be expected. Moreover, as it will be shown next by charge population analysis, there is a charge accumulation around the N atom, which in an initial-state approximation, will induce a core-level-shift towards lower binding energies with respect to N in a graphitic network. To further confirm this assumption we have carried out a set of DFT-based calculations of the N 1s core level binding energy for the following different molecule adsorption configurations with respect to N in a graphitic environment. Our theoretical calculations predict a value of 399.2 eV for the binding energy of the adsorption configuration shown in Fig. 1g, in excellent agreement with the spectrum shown in Fig. 1h. Details about these calculations are shown in the Supplementary Fig. 2, the Supplementary Note 3 and the Supplementary Tables 1 and 2. Interestingly, a binding energy of 399.7 eV has been reported for N 1s strongly interacting with C atoms after azidotrimethylsilane functionalization[40]. An intensity analysis of the C 1s core-level peak is shown in the Supplementary Fig. 3 and the Supplementary Note 4, and more arguments for the assignment of the N 1s core level peak to our molecular-linked group is given in the Supplementary Note 5 and the Supplementary Table 3.

Remarkably, if p-AP is dosed on pristine graphene (no SAVs) no absorption takes place, as depicted by STM and XPS measurements (Fig. 1g). Thus, we propose a mechanisms consisting in an on-surface-induced dehydrogenation reaction at the amino-groups of the p-AP molecule catalysed by the excess of charge at the dangling bonds positions.

We have checked the thermal stability of the anchored molecules, and no significant changes were found neither in the line-shape and intensity of the XPS spectra nor in the STM images appearance up to temperatures above 250 °C, indicating the covalent nature of the bonding, and allowing us to disregard any unspecific molecular adsorption (Supplementary Fig. 4 and Supplementary Note 6). Moreover, we have performed mass spectrometry thermal desorption experiments that show unspecific desorption from the surface takes place at about 60 °C and the absence of any molecular fragment that may indicate rupture of the bound molecule upon annealing (Supplementary Fig. 5 and Supplementary Note 7). It is well known that implanted heteroatoms on the graphene lattice are stable until temperatures of about 1,000 °C (ref. 41). Our stability value, above 250 °C, is similar to that found for aryl functionalized graphene devices[9,21]. This behaviour is also predicted by temperature-assisted DFT-based molecular dynamics calculations on the system, which, at this temperature, maintains its low-temperature structure, giving idea of the high structural integrity, robustness and stability of the formed covalent bonds (Supplementary Note 7).

Interestingly, a recent work explores the catalytic activities promoted by the unsaturated dangling bonds that may remain on a surface after SAV formation[24]. However, those experiments were performed on epitaxial graphene on metals, and it has been shown by DFT calculations that the C atoms at the vacancy edge back-fold towards the metallic substrate promoting bonding with it (ref. 31). This is different in our case, where the existence of a saturated buffer layer decouples the graphene, and strongly reduces the possibilities of back-bond.

**Calculations on the mechanism: STM images and energy barriers.** With all this chemical information in mind we have carried out a large battery of DFT-based calculations to confirm the above-mentioned scenario (see detailed computational details and models in the Theoretical Section of the Supplementary Methods, the Supplementary Notes 1,8–10, and the Supplementary Figs 6 and 7). The most stable structure in terms of total energy minimization is shown in Fig. 1f, which corresponds to the p-AP molecule lying perpendicular on the surface with the doubly dehydrogenated nitrogen atom integrated (with an out-of-plane distance of 0.65 Å) into the graphene lattice within the SAV. Note that a single-substitutional N atom will exhibit a triangular feature very similar to the one we observe but with the threefold irradiating features[25]. To reach this conclusion we have performed calculations of the most likely geometries of a p-AP molecule interacting with the SAV to compute the different formation barriers for different paths. We have evaluated plausible reaction mechanisms when the p-AP approaches the vacancy either by the OH or NH$_2$ groups. We have computed the height of the barrier, $\Delta E$, at the transition state (TS) in the double ($-$NH$_2$) dehydrogenation process of the p-AF molecule. This TS has been investigated within the climbing-image nudge elastic band approach (see full details in the Supplementary Fig. 7 and Supplementary Notes 8 and 9) where the initial, the final, and 12 intermediate image-states were free to fully relax to get a converged minimum energy path (MEP). The result of this TS energy barrier reveals that this double dehydrogenation process of the intact p-AP molecule is efficiently catalysed by the SAV on the surface, yielding values of 1.3 and 0.7 eV for the first and the subsequent second dehydrogenation, respectively, of the amino terminating group, and a total energy gain of 3 eV. In addition, for the case in which a p-AP molecule would approach towards

the SAV via the –OH terminating group we have also calculated the MEP for the surface-induced dehydrogenation of the hydroxyl group. In this case, although the energy barriers are comparable, the net gain of energy in this process is significantly lower as in the mentioned double (–NH$_2$) dehydrogenation (around 1.5 eV), besides the fact that the remaining dehydrogenated oxygen would not be able to perfectly integrate within the SAV into the graphene lattice, only getting attached to one (of three) of the C atoms of the SAV. All calculated energy barriers are deeply discussed in the Supplementary Fig. 7 and the Supplementary Notes 8 and 9.

To gain further insight into the properties of the anchored p-AP molecules we have recorded a series of STM images at different bias voltages and compared them with theoretical STM-imaging calculations (Fig. 2). This figure corresponds to a QFSMG sample and shows enhanced resolution due, very likely, to picking up a single hydrogen atom in the tip apex. Curiously, all atoms of the surface can be seen through the bump, increasing the resolution as we approach the tip. One may wonder where the adsorbed molecule is. To solve this puzzle, we have performed theoretical STM-imaging calculations with a hydrogen-functionalized tungsten tip (see full details in the Supplementary Fig. 6 and the Supplementary Note 10) and we have found a very good agreement with the experimental images. These calculations show that as the bias voltage is increased the atomic resolution and bump intensity smooth out. We experimentally checked that this behaviour is reversible and it is a strong and elegant confirmation of the substitutional nature of the N in the graphene atomic network.

The fact that the molecule is essentially not seen in the STM images can be understood in a qualitative way by the calculated

density of states depicted in Fig. 3. Although the real molecular height is 6.5 Å according to our DFT calculations, the apparent height determined by STM is in the range from 1.0 to 1.6 Å (Supplementary Fig. 1 and Supplementary Note 2). Our images are recorded in occupied states because the doping of the substrate and. therefore, at voltages between −700 mV and the Fermi level we cannot expect any contribution from the molecule (Fig. 3). The unique signal from the molecule contributing to the STM images is the queue of an electronic state related to the N atoms (marked with an * in Fig. 3), which extends around the adsorption area.

This behaviour does not change when H-atom is lost from the tip-apex. The molecule is also visible as a bump (Fig. 1c corresponds to this case). In this case, the well-resolved hexagonal graphene lattice is not shown but the height of the molecular protrusion remains within the same range as it is shown in Supplementary Fig. 8.

**Charge rearrangement upon covalent bonding.** Interestingly, the N atom in the proposed structure has the charge unbalanced, since it is surrounded by four C atoms (Fig. 1f). To evaluate the charge state of the molecule and the local charge on each atom involved in the adsorption we have carried out a Bader analysis on the DFT-optimal adsorption structure (see Theoretical Section of Supplementary Methods for further details). This study shows that a p-AP(-H$_2$) molecule transfers 0.56e$^-$ towards the graphene lattice, while the N atom (considered as belonging to the molecule) accumulates an extra electronic charge of around 0.82e$^-$ with respect to its neutral state, coming from a charge rearrangement in the molecule, leaving the rest of the molecule

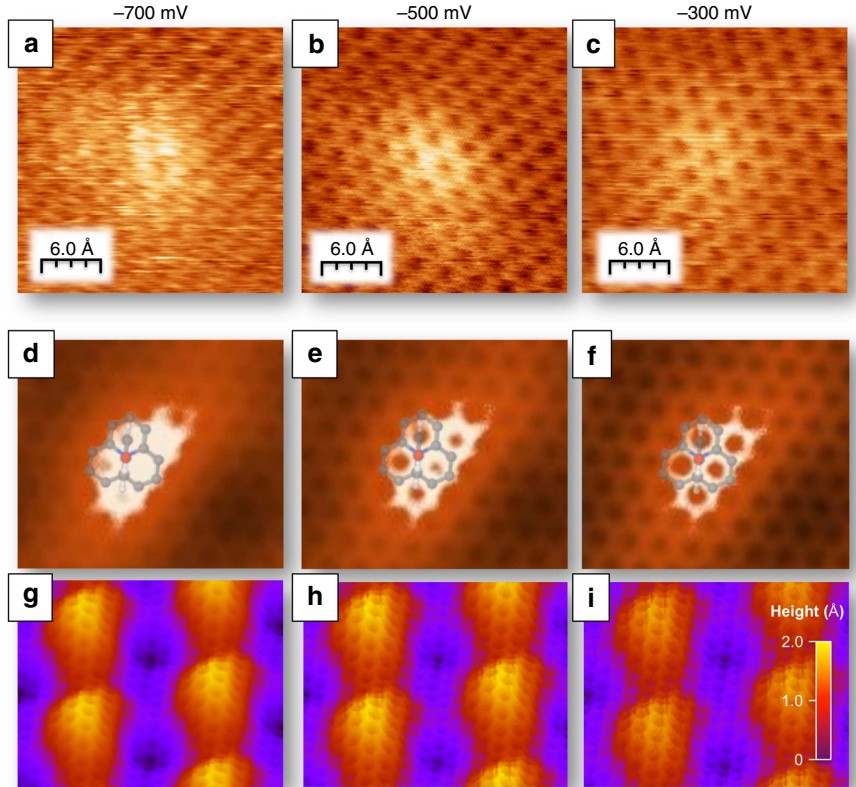

**Figure 2 | The p-AP molecule replaces substitutionaly a carbon atom with the nitrogen.** Series of experimental and theoretical STM images showing the dependence of the image morphological shape with the applied tunnelling bias for V$_s$ = −700, −500 and −300 mV at constant-current regime. (**a**–**c**) Experimental images; (**d**–**f**) Theoretical STM images; (**g**–**i**) Theoretical 3D topographical STM images. The experimental results are well reproduced by theory when a H atom is included on the tip apex (see details in the Supplementary Fig. 6 and the Supplementary Note 10).

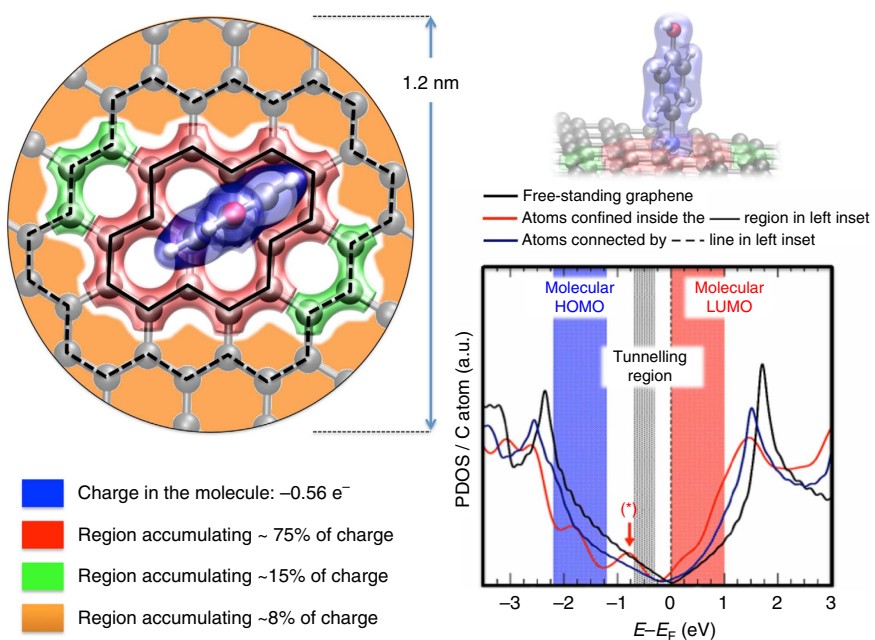

**Figure 3 | Charge redistribution in the region around the molecule upon covalent bonding.** (Left panel) Pictorial sketch showing that most of the net charge provided by the molecule ($\sim$75%) is accumulated and distributed among the nearest C atoms (red shadowed region), $\sim$15% among more distanced C atoms (green shadowed regions) and the rest ($\sim$8%) among the rest of the C atoms of the graphene sheet (orange background). (Right panel) Projected density of state profiles per C atom (PDOS/C atom; in a.u.) versus energy (referred to the Fermi level; in eV) onto different sets of C atoms as we increase the distance from the $N$ atom: (solid-red line) C atoms confined inside the region delimited by the solid-black line in left panel; and (solid-blue line) C atoms connected by the dashed-black line in left panel. For comparison, we also show the density of states per C atom in free-standing graphene (solid-black line). Shadowed in different colours are the molecular HOMO (blue) and LUMO (red) position of the molecule, and the tunnelling region of the experimental images (grey). (*) denotes a N related electronic state on the neighbouring C region.

with a net loss of almost $1.4 e^-$. This is related to the fact that the N atom integrates into the graphene lattice, stabilizing its charge state by adsorbing and accumulating charge on the N atom from the rest of the molecule, and acting as a drain-pipe towards the C atoms at the neighbourhood.

The left panel of Fig. 3 shows that most of the net charge provided by the whole molecule ($\sim$75%) is accumulated and distributed among the nearest C atoms (red shadowed region in Fig. 3), $\sim$15% among more distanced C atoms (green shadowed region in Fig. 3) and the rest ($\sim$8%) among the rest of the C atoms of the graphene sheet (orange background in Fig. 3). Thus, we show that the covalent link of a p-AP on the surface induces an electronic distortion in graphene that remains confined to the covalent bonding position, similarly to other well confined doping impurities[25]. Therefore the neighbourhood of the anchoring position can be regarded as a basin of charge of about 1 nm of radius, coinciding with the extension of the pronounced protrusions in the STM images. This charge redistribution has been theoretically checked by using three different charge population frameworks, with differences between them not beyond a 15% (see the Theoretical Section of Supplementary Methods). Interestingly, this region of charge accumulation around the adsorption site has not being reported for C–C coupling[21].

To evaluate the spatial extension of the electronic charge transferred from the molecule towards the graphene lattice, right panel of Fig. 3 shows a depiction of the projected density of state profiles per C atom (PDOS/C atom) versus energy (referred to the Fermi level) onto different sets of C atoms as we increase the distance from the N atom. We differentiate among C atoms confined inside the region delimited by the solid-black line in left panel of Fig. 3 (solid-red line in right panel of Fig. 3); and C atoms connected by the dashed-black line in left panel of Fig. 3

(solid-red line in right panel of Fig. 3). For comparison, in right panel of Fig. 3 we also show the density of states per C atom in free-standing graphene (solid-black line). It is important to notice that for the PDOS profile for C atoms of the closest-to-$N$ region—within the solid line–, the resembling of pristine graphene DOS is quite poor, and some induced features arises just above and below associated to a reflection of the electronic properties of the positively charged molecule. In particular, the one arising between $-1$ and $0\,eV$—labelled as (*) in right panel of Fig. 3—corresponds to an induced density of states by $N$ on their closest C atoms. On the other hand, the highly hybridized molecular HOMO and LUMO positions are indicated as blue and red shadowed region, respectively, and they have visibly spread out by effect of the chemical interaction. The tunnelling region, where experimental images were recorded ($-0.7/-0.3\,eV$) is also indicated in the right panel of Fig. 3. Interestingly, the tunnelling region does not overlap at all with the molecular HOMO, which explains the lack of STM signal coming from the molecule in the image.

Nevertheless, as we move away from the N atom, the C-atoms set (crown)—between the dot and solid lines in Fig. 3—, more accurately resembles the DOS profile of the pristine graphene, and a noticeable mitigation of the molecular features around the Fermi energy is also evident.

**Control on the density of immobilized molecules.** The density of immobilized groups is an important point when focus on applications[21], and it can be easily quantified by statistical counting in STM images. In the example shown in Fig. 1 this value is about $5.5\times10^{12}\,cm^{-2}$ but it can be controlled by changing the duration of the ion irradiation[41]. Importantly, we verified that four Langmuirs are enough to cover all vacancies

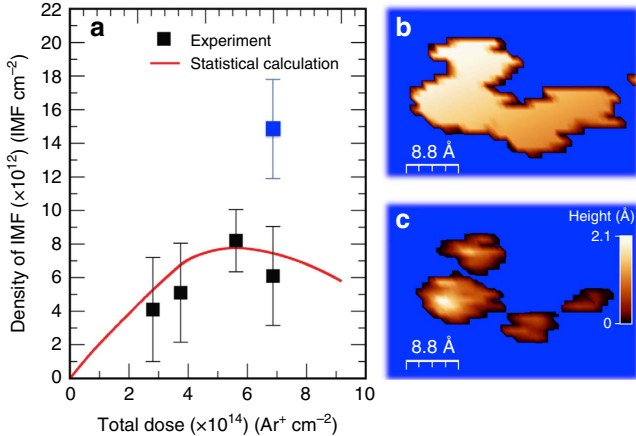

**Figure 4 | Evolution of the covalently linked vacancies with the ion dose.** (**a**) Density of isolated molecular features (IMF), in $\times 10^{12}$ IMF cm$^{-2}$ units, versus total dose of Ar$^+$ ions, in $\times 10^{14}$ Ar$^+$ cm$^{-2}$ units: (black-solid squares) statistical counting extracted from a large amount of experimental STM sessions for different Ar$^+$ irradiation time; (red-solid line) Akima-spline fitting of the statistical predictions extracted from a calculation based on the generation of random circle-shaped events of radius 1 nm within a $(100 \times 100)$ nm$^2$ grid by differentiating isolated from overlapped events (**b**) STM image of several features that have coalesced into an unique one and are indiscernible and (**c**) Same image as (**b**) changing the saturation to see different maxima: the bump is formed by coalesce of four smaller protrusions. If the radius of 1 nm is not taken into consideration the blue point is obtained after counting IMFs. STM image obtained using a bias of $-798.7$ mV and tunnel current of 0.084 nA. The magnitude of the error bars has been taken as the difference between the maximum and minimum values extracted from our phenomenological statistics, for the same STM scanning area, centered at the corresponding average values.

with p-AP molecules, indicating the high throughput of our reaction. Fig. 4 shows the evolution of the density of covalently linked molecules with the irradiation time or dose. To make this figure, independent molecular features (not overlapping) on several STM images were counted for different sputtering time, which is proportional to the flux current (all others experimental parameters were kept constant; Supplementary Fig. 9). Figure 4a shows that we can accurately control the density of immobilized molecules. Interestingly, this curve is not linear but the process goes trough different regimes. For low irradiation times or low doses, the formed number of vacancies behaves strictly linear. However, after a particular ion-dose the probability of an ion to create a vacancy within the region affected by a previous ion is higher, and therefore the possibility of overlapping vacancies increase as the irradiation time, reducing the slope of the curve. In Figs 1g and 2 we have seen that adsorption of a single-aminophenol molecule create electronic distortion into a region of about 1.5 nm of diameter. Finally, in our experiments is evident that after the fourth experimental point a decrease in the density of created vacancies is clearly appreciated. This is due to the coalescence of the generated vacancies. Figure 4b represents a typical STM image with a shape indicating that the basins of charge associated to different molecules have overlapped, and therefore the probability of having diatomic vacancies or electronic interferences is high. Nevertheless, if we count the maxima within the STM image by saturating the colours (higher features in light-orange colour), we observe that the single bump consist of four overlapping features (Fig. 4c). If we count these as four independent features, that is, without taking into account the 1.5 nm width, we obtain the point in Fig. 2 in blue colour, which recovers the linearity of the first points.

This mechanism can be theoretical described with a simple phenomenological calculation based on random numbers. We have computed the density for different fluencies removing

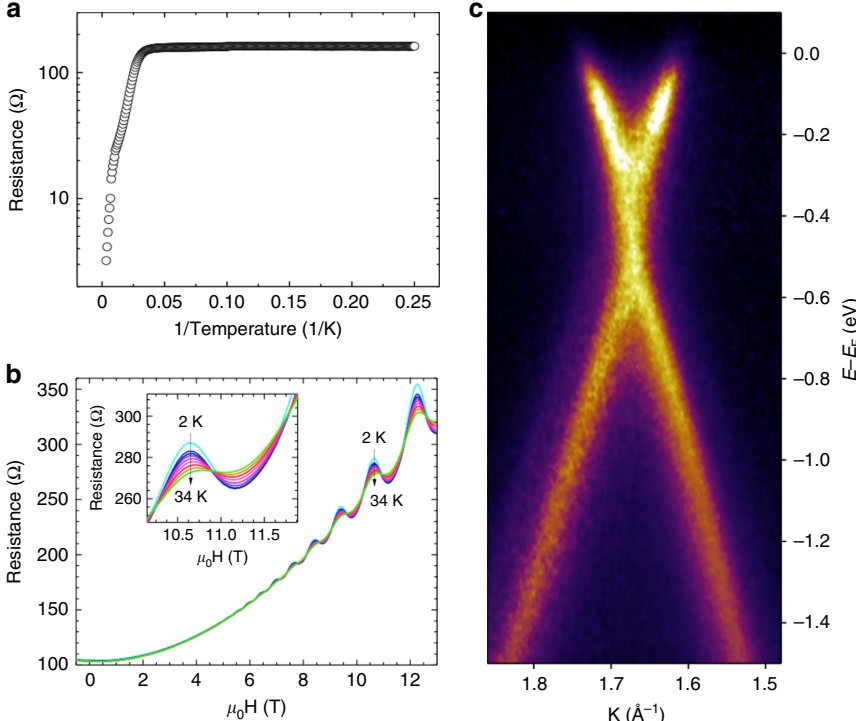

**Figure 5 | The electric and electronic graphene features remain unaltered after p-AP functionalization.** (**a**) Resistance versus inverse temperature at zero field for the p-AP functionalized sample. (**b**) Shubnikov–defHaas oscillations for our functionalized graphene sample on a semi-insulator SiC substrate. (**c**) Photoemission momentum map of a p-AP functionalized SLG. Cut along $k_x$ direction. Photon energy: 34 eV.

from the counting all overlapping events. The overlapping criterion has been taken in such a way that isolated events are considered if a minimum lateral distance larger than 1 nm separates them. This simple computational model reproduces quite perfectly our experimental data, and let us to conclude, both from calculation and experiments, that the maximum number of independent molecules that can be covalently linked is $8 \times 10^{12}$ molecules per $cm^2$. For doses leading to higher coverage many events overlap and the characteristic electronic properties of the system collapse.

It is important to note that assuming a purely geometrical model of hard spheres with the experimental radius of 1 nm, the maximum number of immobilized molecules is about $3.2 \times 10^{13}$ molecules per $cm^2$. The statistical nature of our methodology only allows us to have a density four times lower.

Moreover, from Fig. 1, we can derive the probability of creating a vacancy by an incident ion, because it shall equal the slope of the curve for low doses, and we can derive from our data that this value is about 0.02. Note that the required doses are very similar to that published[24].

**Transport and electronic properties of the functionalized graphene.** Finally, and remarkably, our p-AP functionalization protocol does not substantially alter the electronic properties expected for epitaxial graphene over large macroscopic extensions[39] but it retains the electrical properties of a pristine graphene sample. Fig. 5a shows the resistance ($R$) versus inverse temperature measurements at zero magnetic field recorded for a QSG sample. Of paramount importance is the fact that the functionalized system does not exhibit a measurable gap at low temperatures (below 34 K); the thermally activated regimes at intermediate (between 34 and 140 K) and high temperature (above 140 K) regimes exhibit low-activation energies (see Supplementary Figs 10–12 and Supplementary Note 11 for further details)

Upon cooling below 34 K, $R$ versus magnetic-field (**H**) curves evidence the presence of Landau Levels in p-AP functionalized samples. Shubnikov–de Haas oscillations are evident in Fig. 5b in the 2–34 K range (see inset), indicating that the quality of large graphene domains, despite having been functionalized with organic molecules in a non-meaningless density, is comparable to that of micro-mechanically exfoliated graphene[42].

Moreover, from a fundamental point of view, the electronic band structure of the sample does not seem to be altered (at most slightly modified) after p-AP functionalization. This is shown in Fig. 5c by angle-resolved photoemission (ARPES) measurements of the valence band structure around the K point of the graphene Brillouin zone. Typical linear dispersing π bands of monolayer graphene are seen. The Fermi Level is located above the Dirac point (the crossing point of the π bands) meaning that the sample is n-doped. A gap of about 250 meV is observed between the bottom of the conduction band (300 meV) and the starting point of the valence band. Similar behaviour has been already reported for this type of pristine samples[17]. However, small variation upon adsorption of the Dirac cone has been also reported for epitaxial graphene on SiC functionalized with nanoparticles[38,43]. These results, although unexpected, reflects the confined nature of the interaction. In our case, the charge transferred from the molecule remains confined in the graphene atoms around the adsorption point. The fact that the maximum molecular coverage reached by this protocol is about 1% (Fig. 4) makes the Dirac cone slightly modified upon adsorption, and no relevant shifts can be expected, contrary to what has been previously observed for other adsorbates[14].

Atomic force microscopy (AFM) measurements further prove that this functionalization process did not either alters the morphology of the graphene samples. Topographic and surface potential images of the functionalized samples look very much alike to pristine graphene on SiC, with no evidences of large-scale damage (Supplementary Fig. 13 and Supplementary Note 12).

## Discussion

To validate our mechanism, we have chosen p-AP as a model molecule as it incorporates an aromatic ring and two different ending groups. The only requirement for this methodology is to use an amino-terminated molecule, as the mechanism is highly selective and efficient to this particular group. The other ending group has to be chosen towards potential applications. Several variations of this molecule are nowadays commercially available or possible to synthetize by several laboratory routes. We have experimentally confirmed that our protocol perfectly works also with aminothiophenol and p-Phenylenediamine molecules (Supplementary Fig. 14 and Supplementary Note 13), and theoretically verified that the aromatic nature of the ring is not a requirement. To this aim, we have carried out similar computational charge rearrangement analyses for a large variety of other organic $NH_2$-terminating molecules, with the doubly dehydrogenated N atom incorporated within the graphene lattice (as in the present case): 4-aminothiophenol, 4-aminobenzalde-hyde, 4-(oxiran-2-yl)aniline, among many others, where the same previous behaviour has been also observed. For all these cases the molecules transfer a net electronic charge of around $0.5–0.6e^-$ to the graphene, while N atom accumulates an extra electronic charge of around $0.9–1e^-$. These calculations support the experimental observed generality of the process.

In our methodology the maximum number of independent covalently linked molecules is about 0.25% (Fig. 4). This value, although it seems low, is not so. The important area is the one electronically modified by the p-AP adsorption, which is about $3 nm^2$ per molecule (electronic accumulating regions of Figs 2 and 3). Taking this number into account the optimal coverage is about 50%. We speculate that depending on the targeted application one may like to have a smaller density of immobilized molecules. Thus, for instance, to have independent nanoparticles of about 5 nm of diameter covalently linked to a surface, one has to use densities of about $2–4 \times 10^{12}$ molecules per $cm^2$ (ref. 38). Higher values can be precluded by the physical size of the nanostructure to be immobilized[21]. Indeed, one of the reasons for the unaltered macroscopic properties of graphene is the low density involved, combined with the localized nature of the electronic modifications.

Dehydrogenation and coupling of N takes place exclusively on the vacancies and not in other structural defects. Reactivity enhancement at specific topographical features, as patterning induced by moiré superstructures, has been reported for epitaxial graphene on metal[14,16]. We have confirmed that there is not preferential adsorption or nucleation at step edges, buffer layer-induced reconstruction or other morphological defects. They do not present electronic properties adequate to induce an oxidative coupling of the p-AP molecule. Moreover, we have not found evidences of formation of Stone–Wales or any other structural defects after our soft irradiation protocol, although some high-energy electron experiments (120 MeV) point out that can be formed by continuous irradiation[44]. Calculations indicate that formation of Stone–Wales rearrangements are difficult to produce[45]. In any case, these defects involve uncoupled dangling bonds, and therefore they could be susceptible to induce reaction of the amino groups.

Finally, the highly selective strategy we have presented to covalently linked organic nanostructures into the graphene lattice can be used to provide graphene with new functionalities,

in particular for hybrid constructions[1–3,6,8]. At this point, it is important to mention that alternative efficient functionalization routes in a wet-environment starting from cyano- and fluoro-graphene towards graphene covalently modified with thio-, cyano- or carboxy-groups have been recently proposed[46,47] These strategies lead to development of a broad family of graphene derivatives with organics through consequent chemistries and, together with our present study, enable a selective and controllable way to develop advanced 2D-heterostructures. Among all the potential applications we can envisage those requiring high selectivity, as those related to bio-sensing or bio-field-effect transistors[10,21,27,28]. The redistribution of charge that can be expected upon adsorption on a target immobilized by our protocol could lead to a change in the electric field across the device, inducing changes in the electronic conductivity and the overall device response[20]. Our protocol could be also compatible with other hetero-applications, in which one could combine the graphene properties requiring low-sheet resistance, as solar cells, light-emitting diodes, liquid-crystal displays, touch screens, with other functionalities provided by the new chemical groups. Finally there is a potential use in the fields of spintronics or optoelectronics, as this strategy can be an easy way of introducing elements or magnetic nanotructures to the graphene surface[43,48]. In this sense, we also succeeded to covalently link aminothiophenol in the search of a potential route for anchoring gold nanoparticles that may present optoelectonic properties (Supplementary Fig. 14 and Supplementary Note 13).

In summary, we present an atomistic description of an oxidative coupling reaction of any amino-terminated molecule induced in graphene by the local excess of charge at the C dangling-bonds of SAV generated after mild ion treatment. We have shown that the dehydrogenation of p-AP molecules leads to the substitutional inclusion of the nitrogen into the graphene lattice. This strategy, complementary to other previously reported routes for aryl coupling[8], leads to a selective organic functionalization of graphene without totally disrupting its electronic properties.

## Methods

**Experimental section.** Growth, STM, XPS and ARPES experiments were performed *in situ*, whereas AFM and magneto-resistance measurements were performed extracting the sample out of the UHV system. The STM measurements were carried out in a RT microscope inside of a UHV chamber with base pressure below $2 \times 10^{-10}$ mbar. Images were recorded with negative values of the bias due to semi-insulator character of the substrate. For the high resolution images low currents and a tungsten-tip functionalized with hydrogen atom was used. Vacancies in the graphene lattice were created by sputtering the surface with Argon ions ($Ar^+$) during 45 s at normal incidence angle. The acceleration energy of the electrons inside the ion gun was 140 eV, the current sample was 1 μA and the gas pressure during this process was kept at $1 \times 10^{-7}$ mbar. Aminophenol was dosed from a crucible hold at room temperature inside the UHV chamber.

XPS spectra were recorded using a Phoibos 150 electron analyser equipped with a 2D-DLD detector and an Al-Kα monochromatic X-Ray source. ARPES spectra have been acquired at RT at BaDElPh beamline located at ELETTRA Synchroton in Trieste using a SPECS Phoibos 150 with a 2D-CCD detector system[49].

Resistance as a function of magnetic field and temperature (1.8 K–360 K) was measured on $1 \times 1$ cm$^2$ square samples using the Van der Pauw configuration in 9 T and 14 T Quantum Design PPMS, using Wimbush press-on contacts on Cu-sheet pads placed at the sample corners. The magnetic field was applied perpendicular to the graphene sheet. Resistance measurements were taken in three configurations: two of them optimized for resistivity measurements along nominally orthogonal directions in the sample while the remaining configuration maximized the Hall resistance. Further information is given in the Experimental Section of the Supplementary Methods, Supplementary Figs 10–12 and Supplementary Note 11).

**Theoretical section.** First-principles computational simulations have been carried out within the basis of different DFT-based implementations. In particular, we have combined the localized-basis-set and plane-wave DFT-schemes as implemented in the FIREBALL[50] and QUANTUM ESPRESSO[51] simulation packages, respectively. In the latter, an efficient perturbative van der Waals correction[52] has been adopted

in the calculations to account for dispersion forces and energies. Fine structural optimization of the systems, electronic structure features (such as density of states or Bader analysis of electronic charges[53], as well as climbing-image nudge elastic band approach[54] to compute MEPs and TS energy barriers were calculated by using the QUANTUM-ESPRESSO simulation package, while structural pre-optimization studies, temperature-dependent molecular dynamics calculations and tunnelling currents were calculated by the localized-basis-set code FIREBALL[50]. The transferability of structures and electronic structure results between both DFT-implementations has been fully successful as reported in abundant previous literature by our group. Tunnelling currents for the STM images were calculated by using a Keldysh–Green function formalism, together with the first-principles tight-binding Hamiltonian obtained using the local-orbital DFT-FIREBALL method[50,55]. Within this theoretical STM-imaging approach the scanning tip and the sample are considered and treated separately, which permits the functionalization of pure metal scanning tips to check the influence in the simulated STM images[56]. A fully detailed explanation of the theoretical methods and the models used in the calculations can be found in the Theoretical Section of the Supplementary Methods and the Supplementary Notes 1,8–10.

**Data availability.** The data that support the findings of this study are available from the corresponding author on request.

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

## Acknowledgements

The research leading to these results has received funding from the Spanish MINECO (through Grants No. MAT2014-54231-C4-1-P, MAT2016-80394-R, RYC-2014-16626 and RYC-2015-17730), from the EU via the ERC-Synergy Program (Grant No. ERC-2013-SYG-610256 Nanocosmos) and the European Union Seventh Framework Program (Grant No. 604391 Graphene Flagship), and from the Comunidad Autónoma de Madrid (CAM) via the MAD2D-CM Program (Grant No. S2013/MIT-3007). We also thank the computing resources from CTI-CSIC. R.L. acknowledges financial support from Spanish MINECO under Grant agreement No. CONSOLIDER INGENIO CSD2009-00013.

## Author contributions

R.A.B. and J.A.M.-G. developed the functionalization process. R.A.B. performed most of the characterization experiments, and analysed the data obtained, under the supervision and guidance of M.F.L., M.G.-H., F.M. and J.A.M.-G. J.I.M. made the theoretical calculations and compiled supplementary information. R.L. and F.M. performed the transport measurements. J.M.B. and W.S. prepared epitaxial graphene substrate. N.R.A., I.P. and S.T. performed ARPES measurements, C.M. and F.J.P. made the AFM and XPS measurements, respectively. J.A.M.-G. wrote the main body of the manuscript. All authors discussed the results and commented on the manuscript.

## Additional information

**Competing interests:** The authors declare no competing financial interests.

