## [Peer Review File · Nature Communications]

Reviewers' Comments:

Reviewer #1 (Remarks to the Author)

The manuscript reports about the functionalisation of single layer graphene on SiC with p-aminophenol molecules and its effect on G properties.

Different experimental techniques have been used and extensive detailed theoretical simulations have been performed to explain the experimental data and support the conclusions.

The functionalisation is achieved by creating single vacancies on graphene using low Ar energy ions.

I think that the topic of the paper is potentially interesting and that the paper deserves consideration but I have several concerns which need to be assessed before a final recommendation can be given.

1) The main claim of the paper is that chemisorption of p-aminophenol occurs at single graphene vacancies on SiC without disrupting the highly desired properties of graphene (high mobility of carriers, Dirac cone etc.).

The sentence in the conclusions:

"In our case the phenol-linked molecule can be considered as a chemical spacer or linker for anchoring more complex molecular structures in the search of designing advanced bio-sensors or tailoring the electronic properties of the graphene layer."

is in my opinion too vague and to some extent questionable.

If the electrical properties are maintained and if a gap of 250 meV is observed and a "Similar behaviour has been already reported for this type of pristine samples" then this functionalisation will not be usable for "tailoring the electronic properties of the graphene layer".

Maybe that using these molecules as a linker for gold nanoparticles will provide a route for such general goals but this is not yet proven here.

Moreover, the defects created by ion sputtering are not spatially ordered, and so also the array of nanoparticles which might be obtained in this way will be affected by a similar limitation.

The estimated coverage of p-aminophenol is very low (less than 0.5 %, see supporting information).

Is this concentration high enough for this functionalisation to be effective for some perspective application, and in the wording of the authors, "non meaningless" ?

I think the perspective applications should be better identified and delimited in consideration of the low coverage of molecules attained here.

At which density of adsorbates, the properties of graphene will start to be modified by this kind of functionalisation ? This information is important for the perspective use of this method in applications where a significant amount of adsorbed molecules may be required. The information about what happens at higher coverage is also interesting by itself because at higher coverage the gap width might be larger.

2) It has been recently published (Celasco E. et al, Phys. Chem. Chem. Phys., 2016,18, 18692-18696) that the adsorption of carbon monoxide at single vacancies induced by low energy ion bombardment in graphene is affected substantially by the nature of the substrate. The vacancies of graphene were reported to be chemically inert if the substrate is not reactive. One possible reason may be found in the passivation of the dangling bonds at the vacancy by bonding to the

substrate.

The present study shows, at variance, that, at least for p-aminophenol, vacancies are reactive also for a chemically inert substrate like SiC.

The authors should discuss the possible differences of their system with respect to vacancies drilled on graphene on Cu.

3) The charge transfer balance suggested in fig. 3 is appealing. It should, however, be confirmed by experimental STS curves taken at different distances from the molecule or at least images recorded at different biases should be shown. For example, from the calculations of fig. 3, I would expect a different behaviour at negative and positive bias on atom 1 and no difference on atom 3.

4) In case of tip functionalisation it happens that the contrast changes in subsequent scans over the same area. How does the surface would appear when no hydrogen atom sits on the tip ? Under such conditions the admolecule should become visible in the STM image at least at some voltages.

By the way, which is the bias of the images shown in Fig. 1 ? Is the tip functionalised or not? If the molecule is upstanding, as suggested by the optimised geometry shown in the same figure, the height should be significantly higher than reported. Again it is possible that a bias of 700 meV is too low to be under topographical conditions.

Does AFM provide any information about the height of the molecule?

This information is required to exclude experimentally that the carbon ring of the molecule, rather than only nitrogen, is indeed present at the surface.

5) The C1s spectrum shown in the supplementary file (Fig. S5) only demonstrates that graphene is not affected by the annealing procedure to 250 °C. If the coverage of molecules is the one estimated by the authors since the molecule has 6 C atoms the intensity of the signal due to the molecule should be between 2 and 3 % of the signal to due carbon in graphene. In any case, the C1s spectrum before functionalisation should also be shown in order to make convincing conclusions.

The N1s spectrum demonstrates that N is still present after annealing to 250 °C.

What about the O1s signal ? It should have an intensity larger than the N1s signal. Its presence would prove that all the molecule is still present and not only the N atom.

6) In the supplementary file it is stated that the width and height shown in fig. S6 and used in the text are obtained under "several bias and current conditions".

The use of heights and widths recorded under different bias conditions is at least questionable.

In Fig. S7 it is indeed shown that the width changes significantly with bias while the height does not. Because of this I think it is not correct to include in the same hystograms FWHM recorded at different biases.

It would be more correct to insert in fig. S7 the (statistical) error bars on the width as a function of bias.

Minor points:

a) The inset h in Fig. 1 is misleading. Inside the picture it is written: before and after evaporation, while in the caption and in the text of the paper it is stated that both spectra are recorded after evaporation but in presence and in absence of vacancies, which appears more pertinent.

b) spelling:

dehydrogenation and not dehydrogenation page 1 line 26;

vacancicy page 5 line 126;

via de -OH.. I think: via the -OH.. page 6 line 176;

considered and not consider page 13, line 315;

Reviewer #2 (Remarks to the Author)

The manuscript "Highly selective covalent organic functionalization of epitaxial graphene" written by Bueno et al. reports an opportunity to functionalize epitaxial graphene through covalent bonding of any amino-terminated organic molecule.

Despite of well elaborated UHV STM/AFM data, definitely, the novelty and significance of the present manuscript do not meet the requirements of any Nature family paper. There have been published so many reports describing the covalent functionalization of graphene starting from pristine graphene and also from various graphene derivatives involving graphene oxide, fluorographene etc.

The key challenge of the community is not only to achieve the selective functionalization of graphene but also the possibility to control the degree of covalent functionalization and perform the experiments in a large scale. Definitely, this is not a case of the present study. The authors do not address the quantitative aspects of the functionalization at all.

With a high probability, the covalent functionalization occurs in the very low extend as confirmed (indirectly) through the statements of the authors ("the electronic band structure of the sample does not seem to be altered after p-AP functionalization"). The authors should specify the goal of the study and advantages/drawbacks of the retained electronic properties of graphene after the covalent functionalization. Why this would be exploited e.g. in spintronics as mentioned in the abstract?? Indeed, the scientific community working in the field of graphene needs rather opposite alternatives: controllable functionalization towards tailored graphene properties or controllable 2D chemistry enabling to create applicable 3D superstructures.

Some other points:

- Introduction and literature referencing are absolutely not sufficient. Definitely, there are so many functionalization approaches, which are overlooked (not reported) by the authors.
- The authors have checked the thermal stability of the anchored molecules through STM images and computational chemistry. However, the conclusions should be supported by thermal analyses and XPS (after the thermal treatment).
- The authors should quantify the number of functionalizing groups and nitrogen embedded into the graphene lattice after the dehydrogenation of p-aminophenol molecules. Ideally, they should demonstrate how these contents are controllable.

Taking into account the above mentioned aspects I suggest to reject the manuscript in its current form.

Reviewer #3 (Remarks to the Author)

The authors have demonstrated an interesting way to functionalize epitaxial graphene with organic molecules. The procedure is via the creation of single vacancies by ion irradiation followed by dehydrogenation of p-aminophenol molecules leading to the covalent bonding between the molecule and graphene through nitrogen incorporation into the 2D graphene sheet. Moreover, the characteristic electronic properties of graphene are retained in presence of covalent functionalization. The experimental results are analyzed by ab initio calculations, simulations of STM images and calculations of energy barriers involved in the functionalization process. The work is quite impressive and convincing, which may lead to eventual acceptance of the paper. However, some points have to be clarified before.

1. In the introduction, the importance of aromatic molecular functionalization of graphene has to be stressed. Specifically, the reason for the particular choice of p-AP molecule should be justified.
2. Are the single vacancies only defects those are created during ion irradiation? What about divacancies, Stone-Wales defects etc.?
3. Can the authors show the dependence of coverage of p-AP molecules on retaining the Dirac cone behavior of graphene?
4. The geometries of the system along the minimum energy path should be shown along with the

corresponding description. This should be done for both nitrogen and oxygen incorporation into the graphene lattice.

5. In Fig.2, the comparison between theory and experiment for different bias voltages is not so convincing. The authors need to explain the figure more carefully in order to convince the readers regarding a good agreement between theoretical and experimental images.

Specific answer to reviewers

First of all, I would like to thank all the reviewers by their general positive evaluation of its potential interest. Their criticism, and careful reading of the manuscript, has helped us to produce a more focused and complete version of the manuscript.

Reviewer 1

1)"In our case the phenol-linked molecule can be considered as a chemical spacer or linker for anchoring more complex molecular structures in the search of designing advanced bio-sensors or tailoring the electronic properties of the graphene layer." is in my opinion too vague and to some extent questionable.

We agree that this statement could be too vague and speculative, for this reason we have included a new discussion section, where all these aspects are addressed and referenced, and left the conclusion section more focused. Moreover we have reworded the conclusions to take into account the following specific comments.

General Action taken: we have summarized the conclusions to avoid speculations and included an extended paragraph about possible applications in the new discussion section.

Q1.1.- If the electrical properties are maintained and if a gap of 250 meV is observed and a "Similar behaviour has been already reported for this type of pristine samples" then this functionalization will not be usable for "tailoring the electronic properties of the graphene layer".

We fully agree with the reviewer in this important point. Our methodology is not focused to tailor electronic properties of the graphene layer, but rather to induce other properties that may be of use on different fields, particularly in the search of hybrid configurations. Moreover, in the paragraph where ARPES is discussed, we have added two references to groups that have functionalized epitaxial graphene on SiC and found similar values. We apologize for this comment and we thank the reviewer for the observation. Nevertheless, this pinning of the electronic gap is not against their use as FET sensors, as mobility or conductivity changes can be also meaningful for these applications.

Action taken: the potential and possibilities of the proposed methodology are now deeply commented in the discussion section. The tailoring of the electronic properties has been removed. See also Q1.4.

Q1.2.- Maybe that using these molecules as a linker for gold nanoparticles will provide a route for such general goals but this is not yet proven here.

During the 4 months from the original submission of the manuscript we have been working in this point, and we have succeeded to immobilize gold nanoparticles and to determine by chemical methods the number of free thiols on the sample. These studies are under progress. As an example, we show you a XPS spectrum of the S2p core level peak, where the maximum of the peak corresponds to the thiol group (-SH) and the N1s states at the same energy as fig. 1. However, since this is still an on-going focused experiment, involving many other groups from chemistry, synchrotron radiation and AFM, exceeding the main goal of this article, we include a new section in the ESI accounting for the XPS characterization of the peaks.

Action taken: we have included a new section (numbered as 11) on the ESI discussing the S2p core-level peak when aminothiophenol molecules are used.

Moreover, the defects created by ion sputtering are not spatially ordered, and so also the array of nanoparticles, which might be obtained in this way will be affected by a similar limitation.

Actually, we do not feel this point as a limitation of the methodology since we are able to control the surface coverage (see next and new section devoted to this topic). It is not our main goal to produce a kind of a pattern, but a global surface modification. We have included in the discussion references to pattern surfaces with coincidence networks (known as moiré superstructures) in which the authors see some effect. It is also true that our methodology can be performed using focused ion-guns to draw molecular modified paths on the graphene surface, that are going to be functionalized. However, we have not studied this point.

Q1.3.- The estimated coverage of p-aminophenol is very low (less than 0.5 %, see supporting information). Is this concentration high enough for this functionalization to be effective for some perspective application, and in the wording of the authors, "non meaningless"?

And also:

Q1.5.- At which density of adsorbates, the properties of graphene will start to be modified by this kind of functionalization ? This information is important for the perspective use of this method in applications where a significant amount of adsorbed molecules may be required. The information about what happens at higher coverage is also interesting by itself because at higher coverage the gap width might be larger.

We will reply to these questions on the same following paragraph.

This is a very important point raised by reviewers 1 and 2 (coverage dependence and its implications). We have performed herein new experiments showing that the coverage can be scaled easily with our methodology by controlling the irradiation time. New Fig. 3 (see also

hereafter) shows the evolution of the density of covalently linked molecules with the irradiation time or dose. Controlling the sputtering time, which is proportional to the flux current, we can accurately control the density of immobilized molecules, which can be directly obtained by inspection of STM images (see a series of STM images in section 10 of the ESI). For low irradiation times, the dependence is strictly linear, however, after a particular dose the probability of an ion to create a vacancy within the basin of charge of a neighbouring one turns into high, and therefore the possibility of overlapping increase as the irradiation time. In our experiments this is evident after the fourth experimental point, where a decrease in the density is clearly appreciated. The right part of this figure represents typical STM images with a shape indicating that the basins of charge have overlapped, and therefore the probability of having diatomic vacancies or electronic interferences is high. Nevertheless, if we count the maxima within the electronic signature at the surface (higher features in blue colour) the linearity is recovered. From this figure we can conclude that the maximum number of independent molecules that can be covalently linked is 8×10^{12} molec/cm². It is important to notice that assuming a model of hard spheres with the experimental radius of 1 nm, the maximum number of immobilized molecules is about 3.2×10^{13} molec/cm². The statistical nature of our methodology only allows us to have a density 4 times lower.

Nevertheless, reviewer 1 has well estimated the maximum coverage. In our methodology the maximum number of independent covalently linked molecules is about 0.025 %. This value, although seems low it is not, since the important area is the affected by the electronic basins, which is about 3 nm² per molecule. Taking this number into account the optimal coverage is about 50%. We speculate that, depending on the targeted application, one may like to have smaller number, for instance, to have independent NP that are about 5 nm of diameter, one probably has to use densities of about $2-4 \times 10^{12}$ molec/cm². Precisely, one of the reasons for the unaltered macroscopic properties of graphene is these low densities, combined with the localized electronic modifications.

Moreover, this mechanism can be theoretical described with a simple phenomenological calculation based on random numbers. We have computed the density for different fluencies removing from the density all overlapping events. The overlapping criterion has been taken in such a way that isolated events are considered if a minimum lateral distance larger than 0.1

nm separates them. This simple computational model reproduces quite perfectly our experimental data, and let us to conclude, both from calculation and experiments, that the maximum number of independent molecules that can be covalently linked is around 8×10^{12} molec/cm². For doses leading to higher coverage many events overlap and the characteristic electronic properties of the system collapse.

Summarizing,

Actions taken: we have included new experiments, and a new section in the main text: "Control on the density of immobilized molecules", as well as a new section in the ESI (section 10). This new section includes a new calculation and a new figure of the density of linked molecules, indicating that: i) the coverage can be easily regulated; ii) the maximum coverage we may have is statistically limited to 8×10^{12} molec/cm²; and iii) these low numbers are not a problem but may be an advantage for immobilizing larger nano-objects where steric impeachment has to be controlled.

Q1.4.- I think the perspective applications should be better identified and delimited in consideration of the low coverage of molecules attained here.

We fully agree with this appreciation and we have made an important effort to identify them and explain which could be the implications of this research.

Graphene is highly inert towards chemical reactions with most materials or species. Functionalization is regarded as a way of providing graphene with novel and emerging functionalities, in a hybrid construction. We envisage applications in different areas. In particular, this rational and controlled route for covalent immobilization can be used in several technological fields. Some of them have been proposed in several reviews. We highlight among them the following fields:

Sensors and biosensors: following this strategy, biomolecules can be linked to the graphene surface; opening the door to biosensing, enzymatic biosensing, DNA sensing, bio-field-effect transistors, or immunosensing. In this way, we would like to emphasise their usage within *state-of-the-art* graphene field effect transistors (GFET). This kind of device consists in a graphene channel between two electrodes, with a gate contact underneath to modulate the electronic response of the channel. The surface of the graphene-FET device is functionalized by binding probe molecules for the specific targeted functionalities. When a target molecule binds the molecular receptor on the graphene surface, there is a redistribution of electronic charge leading to a change in the electric field across the device, which changes the electronic conductivity and the overall device response and performance. Similar devices have been fabricated with silicon FETs for years, but achieved limited sensitivity and selectivity.

Spintronics or optoelectronics: this strategy can be used as an easy way of introducing elements or magnetic nanostructures within the graphene surface. This protocol is also compatible with other hetero-applications, in which one could combine the interesting graphene properties requiring low-sheet resistance, as solar cells, light-emitting diodes (LED), liquid-crystal displays (LCD), touch screens, with other functionalities provided by the new groups.

Action taken: a detailed summary of previous paragraphs, including new references, is provided in the discussion section.

Q1.6.- At which density of adsorbates, the properties of graphene will start to be modified by this kind of functionalization? This information is important for the perspective use of this method in applications where a significant amount of adsorbed molecules may be required. The information about what happens at higher coverage is also interesting by itself because at higher coverage the gap width might be larger.

This is a very important point and we thank the reviewer for raising it. This issue has been addressed in the response to the previous point. We have found that exists a maximum number of p-AP molecules that can ensemble independently. Larger densities result in a collapse of the electronic basins of charge of every immobilized molecule or into the formation of unrulred vacancy dimers (or larger). The number of independent evens is in our case 8×10^{12} molec/cm². This value has been both calculated and experimentally determined (see previous figure). To calculate this number we have performed a statistical calculation consisting in estimating the probability of an event to occur inside the radius of a previous event. The calculated curve matches perfectly the experimental data points.

Action taken: It has been already discussed within Q1.3 and Q1.5

Q2.- ...” The present study shows, at variance, that, at least for p-aminophenol, vacancies are reactive also for a chemically inert substrate like SiC.”... The authors should discuss the possible differences of their system with respect to vacancies drilled on graphene on Cu.

We do apologize for not including this very recent reference in our manuscript. Indeed it suggests the possibility of using unsaturated dangling bonds to promote chemical reactions. Unfortunately, in the work of Celasco *et al.* their protocol (very similar to ours) is not working as expected. Two may be the reasons. First, in the Celasco work, a metallic substrate has been used, and it has been shown by DFT calculations that the C atoms at the vacancy edge back-fold towards the substrate promoting the bonding with it[32]. This is different in our case, where the presence of a saturated buffer-layer and intercalated H on the SiC reduce the possibilities of back-bonding. Second, it may be that CO is not the best candidate, since this kind of reactions may be quite selective and CO is a very stable molecule.

Actions taken: 1.- This reference has been included in several parts of the manuscript, as an example of vacancies properties, and as first suggestion of the catalytic role of unsaturated dangling bonds. 2.- A new paragraph discussing the differences induced by the substrate has been included in pag. 7.

Q3.- The charge transfer balance suggested in fig. 3 is appealing. It should, however, be confirmed by experimental STS curves taken at different distances from the molecule or at least images recorded at different biases should be shown. For example, from the calculations of fig. 3, I would expect a different behaviour at negative and positive bias on atom 1 and no difference on atom 3.

We fully agree that one of the most interesting points raised by the theoretical simulations is the charge balance, and its localized nature upon absorption. STS experiments would be useful for a better assignment of the experimental features to the calculated ones. Unfortunately, we do not dispose of a LT-STs microscope where to perform these measurements. Nevertheless, in this particular case, we have calculated the whole STM image, see fig. 2 d-f, where all matrix elements as well as a functionalized tip are included and there are no significant differences at these particular atoms. What we have indeed measured is the

bias dependence. A hint of the bias dependence is given in the figure 2 a-c, and the correspondence with the occupied states calculations are coming through the good correspondence with the STM calculated images. In the ESI (section 8) these values are discussed. With respect to the sign of the bias, as the substrate is SiC we have used doped SiC, and exclusively negative biased images can be recorded.

Additionally, in order to check the validity of the results obtained from the DFT-based calculations of the charge transfer between the molecule and the graphene, as well as the rearrangement of electronic charge after incorporation of the molecules, we have carried out a battery of calculations of the charge transfer within different theoretical approaches. It is well-known that in some cases, depending on the nature and character of the atomic system under study, these methodologies to compute the electronic charge rearrangement may exhibit some kind of arbitrariness, being highly sensitive, for instance, to the localized-basis-sets employed (in the case of localized-basis-set and real-space atomistic simulation packages) or to the expansion of the electronic wave-functions in plane-waves (in the case of plane-wave atomistic simulation codes). For that reason we have used three different methods to compute the atomic charges once the ground-state of the system has been categorically established: i) the Bader population analysis [Bader, R. F. W. in *Atoms in molecules - A quantum theory* (ed Bader, R. F. W.) Ch. 2, 13-49 (Oxford University Press, 1990)] implemented in the QUANTUM ESPRESSO code [QE]; ii) the Mulliken population analysis [R. S. Mulliken, *Electronic Population Analysis on LCAO–MO Molecular Wave Functions. I*, JCP 23 (1955)] implemented in the QUANTUM ESPRESSO code [QE]; and iii) the Löwdin population analysis [(a) Löwdin, P. O. JCP 1950, 18, 365.; (b) Löwdin, P. O. *Adv Quantum Chem* 1970, 5, 185] implemented in the localized-basis-set Fireball code. Interestingly, the results obtained with the three different charge population frameworks do not show significant differences, not beyond a 15%, w.r.t. the values given in the main text – obtained within the approach (i) – which, from the theoretical point of view, reinforces the validity of the results shown in the main text.

Action taken: a sentence has been included in the main text to indicate that different charge population calculations have been performed using different formalisms with the aim to check the charge values, and results agree within a 15% (pag. 11). A new paragraph for discussing the electronic structure in reply to Q4.3, has been included. A mistake that may mislead the reader has been found in old figure 4, where “providing” has been replaced by “region accumulating”.

Q4.1- In case of tip functionalisation it happens that the contrast changes in subsequent scans over the same area. How does the surface would appear when no hydrogen atom sits on the tip ? Under such conditions the admolecule should become visible in the STM image at least at some voltages.

The referee is right in this point, when the tip lose the H-atom, the well-resolved atomic resolution is lost, and the molecule is seen as an unresolved bump, and atomic resolution on the substrate is hardly seen. Figure 1c was recorded without any tip functionalization, and the network that is seen corresponds to the SiC surface reconstruction. As the referee indicates, in this case the molecule is observed and the well-resolved atomic resolution on the graphene surface is lost. Curiously, the molecular apparent height is not significantly changing when comparing images of the molecules with functionalized tip or without (within the error bars of fig. S7).

Action taken: a new paragraph has been included to clarify the low effect of the loss of the H on the estimation of the apparent molecular height. See pag. 6.

Q4.2.- By the way, which is the bias of the images shown in Fig. 1? Is the tip functionalised or not?

Images 1a and 1b were recorded with a H-terminated tip, whereas 1c, with a standard W tip. Bias and tunnelling currents are -511 mV/I=0.36nA; -289 mV/2nA; -1240mV/0.09nA for figures 1a,b and c; respectively.

Action taken: figure caption 1 has been amended, and these values included.

Q4.3 If the molecule is upstanding, as suggested by the optimised geometry shown in the same figure, the height should be significantly higher than reported. Again it is possible that a bias of 700 meV is too low to be under topographical conditions.

This is a very interesting point that merits to be discussed into the text. It is true that in principle our apparent height is about 1.6-2 Å, value much lower than the expected from the structure derived by DFT calculations, which is about 6.5 Å. We are working at about 0.05 nAmps, and, therefore, the estimated tip-surface distance is of the order of the molecule height i.e., the tip has not to overcome the whole molecule when scanning, and then, topographic effects are drastically reduced. However, there are two important points to be taken into account. First, that indeed -700 meV is a high voltage to really tunnel through the HOMO orbital, which, after our calculations, starts to extend below -1.2 V from the Fermi level. Thus, our STM images are formed taking into account tunnelling from graphene and a small queue of a N induced state in the graphene, and not from the HOMO orbital. Looking the PDOS curves, indeed our STM image is formed by graphene contribution exclusively. Another way to qualitative understand this reduction of the apparent height is taking into account the charge depletion at the molecule, and therefore a reduction of the apparent height can be expected, in favour of an enlargement of the molecular feature at the STM (region close to the adsorption point: basins of charge). This is indeed what we observe. Besides, moved by this observation we have performed calculations on the expected theoretical height of the molecule with respect to the carbon corrugation in graphene. Although we are aware that these calculations have to be taken cautiously, they indicate that we could expect apparent ranging randomly between 1.7 and 1.9 Å, varying the applied bias from -700 mV to -300 mV. These values have been obtained by theoretical STM scanning lines along fix trajectories. See ESI, section 8; where they are now deeply discussed and shown.

Action taken: a new paragraph has been included in page 10 to discuss the origin of the reduced height in STM images. Theoretical estimation of apparent heights has been calculated and included in the ESI, section 8. New figure 3 including better description of the PDOS, and a better explanation in the text.

Q4.4 Does AFM provide any information about the height of the molecule? This information is required to exclude experimentally that the carbon ring of the molecule, rather than only nitrogen, is indeed present at the surface.

AFM in air is very difficult when the objective is to find out molecular objects. Laterally the resolution is always limited by the size of the tip, and several artefacts can vary the height in values up to 50%. Here follows an AFM image of 200×200 nm heights ranging from 6.5 to 9 Å high and around 25 Å wide. Our immobilized p-AP molecules are expected to be about 6.5 Å

high, and therefore these bumps could correspond to the immobilized molecules enlarged by the tip-width.

Action taken: due to the low quality of the image we will not include it in the manuscript, and we show it exclusively for review purposes. A paragraph has been included in the section 9 of the ESI.

Q.5.- *The C1s spectrum shown in the supplementary file (Fig. S5) only demonstrates that graphene is not affected by the annealing procedure to 250 °C. If the coverage of molecules is the one estimated by the authors since the molecule has 6 C atoms the intensity of the signal due to the molecule should be between 2 and 3 % of the signal to due carbon in graphene. In any case, the C1s spectrum before functionalization should also be shown in order to make convincing conclusions.*

This is not an easy task, since our substrate is SiC and, therefore, there is a large contribution of bulk C atoms into the C1s core-level peak. Nevertheless, we have made the fit indicated by the reviewer, which is now shown in a new section of the supplementary information.

The next shows the C1s core level peak before and after functionalization. This peak is quite complex, as it contains components about the called buffer layer of SiC and bulk SiC. The fit using three components is quite standard in the literature [S32]. The component at 285.57 eV can be assigned to bulk SiC, whereas the component at 284.78 to the C from the buffer layer. Finally the component at 283.74 corresponds to C sp².

The component of sp² shall increase its intensity after functionalization with respect to the other components, as the C in the aminophenol ring is sp² and the process does not affect the bulk components. This is indeed what happens. From the figure we measured the sp² area normalized to the C from buffer and bulk SiC and we obtained values of 0.43 and 0.51, for clean and functionalized surface, respectively. This variation is about 20%, number that may seem large with respect to our low coverage, but one has to take into account that we estimate

a ratio rather than an absolute value. This number considers both the increase of the signal due to the adsorbed molecules and also the electron attenuation for the signal of the bulk component.

Figure S11. Core level peak of C1s from the clean and functionalized surface. The spectrum is fitted using three peaks associated to bulk SiC (green line), carbon sp² configuration (blue line) and SiC buffer (purple line) in increasing order of binding energy. The sp² area in functionalised surface increased respect to bulk SiC and SiC buffer areas due to the inclusion of the benzene ring of the p-aminophenol molecules.

Action taken: a new section, numbered as 12, including a new figure has been included in the version.

The N1s spectrum demonstrates that N is still present after annealing to 250 °C. What about the O1s signal? It should have an intensity larger than the N1s signal. Its presence would prove that all the molecule is still present and not only the N atom.

The reviewer is completely right in this point. However, and unfortunately, the complex sample preparation of this mm-sized epitaxially grown graphene on SiC samples by CVD-process, although leading to the best-quality reported epitaxial graphene in terms of conductivity and electronic properties (see figure 4a and b, and references 15-17; fabricated at ITME), leads to some oxidation on the SiC substrate. This oxidation has been reported previously, and it is not on the graphene, as shown by Raman, LEED etc... The presence of this peak masks any contribution coming from the oxygen from the molecule. However, when we repeated the experiment with aminothiophenol, that includes a SH group instead of OH, we can follow the presence of the thiol group on the surface.

Action taken: to avoid entering in this discussion, we prefer not to show the O1s peak, as it is not providing any useful information in this case. Nevertheless, a new section devoted to the study of the terminal group of the similar aminothiophenol molecule, S1s core-level shift is

included in the ESI, section 11.

Q6.-in the supplementary file it is stated that the width and height shown in fig. S6 and used in the text are obtained under "several bias and current conditions". The use of heights and widths recorded under different bias conditions is at least questionable. In Fig. S7 it is indeed shown that the width changes significantly with bias while the height does not. Because of this I think it is not correct to include in the same histograms FWHM recorded at different biases.

It would be more correct to insert in fig. S7 the (statistical) error bars on the width as a function of bias.

Completely agreed. However, after new statistical counting, we observed that the variation of the width with the bias was visible because the data were not associated to the error bars. When error bars are included, in new figure S7, this effect is not visible anymore. We apologize by this mistake.

Action taken: these figures and histograms have been looked at again, and error bars on the apparent widths are included.

Minor points:

a) The inset h in Fig. 1 is misleading....

It has been corrected

b) spelling....

They have been corrected.

We really appreciate how deep you went into the manuscript details. We are really grateful.

Reviewer 2

Q1.-Introduction and literature referencing are absolutely not sufficient. Definitely, there are so many functionalization approaches, which are overlooked (not reported) by the authors.

We agree with the reviewer in this point but, at the same time, we partially disagree. We apologize because we have not been able to pass correctly the message (aim) and the submitted version was not well discussed into the light of the existing literature. It is completely true that there have been many functionalization strategies in graphene leading to covalent bonded molecular structures, and therefore there are many reviews around. We have cited some of them that cover most of the work in the field, but unfortunately we cannot cite all, as this will make very dense the references section. However, we have made an important effort to discuss our results in the framework of the previous works. In this sense we have added 3 new reviews that show complementary functionalization routes; 5 references to functionalization strategies and characterization of the graphene surfaces (some of them on SiC substrates); 3 references of functionalization in UHV; 3 more references about the role and formation of the vacancies and finally other 4 about application of functionalized graphene. Most of our results are in the present version compared to that previously published works.

However, and importantly, herein we report more than another functionalization protocol: Our differential point is that we show a novel chemical mechanism, described at an atomic level, by which the excess of charge located in unsaturated dangling bonds of C atoms around single atomic vacancies (SAV) induce a dehydrogenation reaction of any molecule containing an amino group, in which the N gets perfectly incorporated into de graphene lattice, behaving as a C atom in the sense that redistribute the charge of the molecule in the neighbourhood of the vacancy. This unprecedented reported mechanism can be used as a rational functionalization strategy to covalently couple amino-terminated organic molecules, as p-aminophenol (p-AP), to the surface of graphene epitaxially grown on SiC without disrupting its amazing electrical properties over mm distances. In this sense this is also the first time that is reported a full immobilization of organics without altering the most important properties of graphene. As far as we know, no other attempt to measure variation of Shubnikov-de Haas oscillations upon functionalization has been achieved.

Among all the strategies up to now developed to covalently functionalize in ultra high vacuum (UHV) i.e., in highly controlled systems, we would like to comment on a recently published work [ref. 16,Navarro, NanoLett 2016]. In this study they report on the functionalization of epitaxial graphene grown on a metal surface by radical coupling by an uncontrolled excitation of electronic processes. Here we present a deeper study: first fundamental, (as we describe all the confinement of the charge around the molecular anchoring points); second, universal (as our mechanism is valid for any amino terminated molecule); and third, susceptible to applications (as we performed on epitaxial graphene on SiC and we show the prevalence of the functionalization in air, as well as the prevalence of conductivity and Shubnikov-de Haas oscillations)

If we compare with wet chemistry protocols, Haddon's group is one of the pioneers in the field and they have mainly develop aryl -coupling using also a large battery of experimental techniques and theoretical methods. Our results are comparable to them and complementary. We open a new reaction scheme for covalent coupling and we perform the experiments in ultra high vacuum.

In definitive these are the main reasons why we trust that our depicted atomistic mechanism of oxidative coupling leads to a highly selective and rational route for covalent linking of organic molecules, that is alternative to other published routes and also merits publication in a high impact journal.

Action taken: we have tried to better explain the focus and relevance of our work (new paragraph in the introduction section). Moreover, many new references to the functionalization, particularly in UHV have been included. Our results and obtained values are now discussed in the light of the previous literature.

Q2.- The authors have checked the thermal stability of the anchored molecules through STM images and computational chemistry. However, the conclusions should be supported by thermal analyses and XPS (after the thermal treatment).

We fully agree with the referee in this point and we appreciate the comment. Indeed we have performed vast XPS, TPD and STM experiments at different temperatures that we describe in the ESI information, section 7. We have submitted the p-AP functionalized surface to temperature up to 500°C, following the evolution by different experimental techniques. The easiest one is TPD, as we do not have seen any signal coming from the molecule and, therefore, we can conclude that in this range of temperatures there is no desorption or rupture of the linked molecule. This is what one may expect from a covalently linked molecule. STM images recorded after annealing the sample are also fully undistinguishable with respect to that recorded before annealing. Finally, to obtain information about the stability of the chemical bond between the graphene network and the molecules at the vacancy sites, as well as to evaluate the degradation of the p-aminophenol by thermal treatments, we performed X-ray photoelectron spectroscopy (XPS) before and after heating at 500°C, without appreciating any change in the core level lineshape.

The following figure shows the N1s (C1s directly on ESI) evolution with temperature. The curves reveal that the spectral shape, before and after annealing, do not experience any substantial change. This result confirms the thermal stability of the bonds between molecules and graphene network up to 500°C. Additionally, it evidences that the molecules do not undergo degradation processes up to 500°C, indicating the covalent bonding of the layer.

Action taken: in section 7 of the ESI (“Thermal stability”) we have detailed discussed all these points. Nevertheless, we prefer to mention in the text that our molecules are stable up to 250°C, value which is more than enough for many applications and to certain our covalent linking, because for 500°C some degradation of the H interstitial in the QFSG layers has been observed.

Q3.- The authors should quantify the number of functionalizing groups and nitrogen embedded into the graphene lattice after the dehydrogenation of p-aminophenol molecules. Ideally, they should demonstrate how these contents are controllable.

This is a very important point raised also by reviewer 1 (coverage dependence and its implications, see Q1.5 of referee 1).

We have performed here new experiments showing that the coverage can be scaled easily with our methodology by controlling the irradiation time. New Fig. 4 shows the evolution of the density of covalently linked molecules with the irradiation time or dose. Controlling the sputtering time, which in our case is proportional to the flux current, we can accurately control the density of immobilized molecules, which can be directly obtained by inspection of STM images (see a series of STM images in the ESI, section 10). For low irradiation times, the dependence is strictly linear, however, after a particular dose the probability of an ion to create a vacancy within the basin of charge of a neighbouring one is high, and therefore the possibility of overlapping increase as the irradiation time. In our experiments this is evident after the fourth experimental point, where a decrease in the density is clearly appreciated. The right part of this figure represents typical STM images with a shape indicating that the basins of charge have overlapped, and therefore the probability of having diatomic vacancies or electronic interferences is high. Nevertheless, if we count the maxima within the electronic signature at the surface (higher features in blue colour) the linearity is recovered. From this figure we can conclude that the maximum number of independent molecules that can be covalently linked is 8×10^{12} molec/cm². It is important to note that assuming a model of hard spheres with the experimental radius of 1 nm, the maximum number of immobilized molecules is about 3.2×10^{13} molec/cm². The statistical nature of our methodology only allows us to have a density 4 times lower. Moreover, This mechanism can be theoretical described with a simple phenomenological calculation based on random numbers. We have computed the density for different fluencies removing from the density all overlapping events. The overlapping criterion has been taken in such a way that isolated events are considered if a minimum lateral distance larger than 0.1 nm separates them. This simple computational model reproduces quite perfectly our experimental data, and let us to conclude, both from calculation and experiments, that the maximum number of independent molecules that can be covalently linked is 8×10^{12} molecules/cm². For doses leading to higher coverage many events overlap and the characteristic electronic properties of the system collapse.

Actions taken: we have included new experiments and a new section in the main text: “Control on the density of immobilized molecules”, as well as a new section of the ESI (section 10). The new section includes a new calculation and a new figure of the density of linked molecules, indicating that: i) the coverage can be easily regulated; ii) the maximum coverage

we may have is statistically limited to 8×10^{12} molec/cm²; and iii) this low numbers are not a problem but may be an advantage for immobilizing larger nano-objects where steric impeachment has to be controlled.

Taking into account the above-mentioned aspects I suggest to reject the manuscript in its current form.

We think that the deep revision carried out on the manuscript – carefully considering all the aspects raised by reviewer 2 (and by the rest of reviewers) – has significantly helped us to resubmit an enriched revised version where all the new technical aspects and discussions have completed and improved the description of the dangling-bond induced selective oxidation of an amino group, leading to an unprecedented rational and highly selective strategy to functionalize graphene maintaining their outstanding electronic properties. We apologize for the deficiency of our previous discussion and shortage of references to other works. On this basis, we have made a battery of changes to better discuss our results at the light of the current literature. We hope to have convinced reviewer 2 that our revised and more focused version merits now publication in its current form.

Reviewer 3

Q1. In the introduction, the importance of aromatic molecular functionalization of graphene has to be stressed. Specifically, the reason for the particular choice of p-AP molecule should be justified.

The reviewer is correct; and also reviewers 1 and 2 remarked this point. Therefore, in the introduction section we have explicitly included a new paragraph stressing the role of the functionalization as well the justification for this particular molecule. In fact, to validate our mechanism, we have chosen p-AP as a model molecule as it incorporates an aromatic ring and two different ending groups. The only requirement for our methodology is to use a molecule terminated in an amino group, as the mechanism we will describe hereafter is selective to this particular group. The terminal group is the one that has to be chosen paying attention to the possible applications. In our case the phenol-linked molecule can be considered as a chemical spacer or linker for anchoring more complex molecular nanoarchitectures in the search of designing functional structures. Moreover, several variations of this molecule are nowadays commercially available or possible to synthesize by several laboratory routes. We have experimentally confirmed that our protocol perfectly works also with amiothiophenol molecules and p-phenylenediamine molecules, confirming the general validity of this new route. Also, following the recommendation of R1 (Q1.4) we have included at the new discussion section some potential applications for the methodology, as well as new references.

Action taken: a new paragraph explaining this point has been included in the introduction section. Some potential applications are also proposed in the discussion section and, finally, new references have been added.

Q2. Are the single vacancies only defects those are created during ion irradiation? What about divacancies, Stone-Wales defects etc.?

This is a very interesting point deserving further discussion. First, we have only found “catalytic” activity on the vacancies. Here we show some STM images showing that in the steps there are not preferential adsorption or nucleation, meaning that they do not present electronic properties to induce an oxidative coupling of the p-AP molecule.

Left hand side image 100.0nm x 100.0nm, 0,033 nA, -545 mV. Right hand side image 50.0nm x 50.0nm, 0,084 nA, -798 mV

Here we show two STM images of the dosed surface in which terraces separated by steps and other defects (1 layer depressions: stacking-fault type) are visible. The high structural perfection of the samples make that we have to look for them along the sample, as there are not so many. The images show that there is no adsorption on step edges or these other geometrical features, indicating that covalent link is exclusively produced on the previously created vacancies.

On the other hand we have found the following kinetic transition states calculations, indicating that formation of Stone–Wales rearrangements are difficult to produce (see figure below). We have not found evidences of formation of Stone-Wales or any other structural defects after our “soft-sputtering” protocol, although some high-energy electron experiments (120 MeV) point out that can be formed by continuous irradiation [Hashimoto et al., Nature 430, 870 (2004)]. Calculations indicate that formation of Stone–Wales rearrangements are difficult to produce [Fan, et al., PRL 90, 145501 (2003)]. In any case, these defects involve uncoupled dangling bonds, and therefore they could be susceptible to induce reaction of the amino groups. Unfortunately, as we do not have found any, we cannot confirm this suggestion.

Image taken from web page

<http://www.kintechlab.com/solutions/nanotechnology/nucleation-and-growth-of-carbon-nanostructures/>

Action taken: a new paragraph has been included in the discussion section.

Q3. Can the authors show the dependence of coverage of p-AP molecules on retaining the Dirac cone behaviour of graphene?

We thank the referee for giving the value it deserves to this important issue. Unfortunately, we cannot answer this question. The experiment we show is a synchrotron radiation experiment and unfortunately, we only succeeded during the beamtime to get the final coverage sample recorded. Nevertheless, the fact that there are not changes with respect to the clean (no functionalized) sample, indicates that for the maximum amount of coverage can be reached (see new figure 2) the Dirac cone remains unaltered. These results, although unexpected, reflects the confined nature of the interaction. In our case, the charge transferred from the molecule remains confined in the region close to the neighbouring graphene atoms (extended within a region of around 3 nm²). The fact that the maximum coverage we can

immobilize by this protocol is about 1% (see figure 4) makes that the Dirac cone is slightly modified upon adsorption, and no relevant shifts can be expected, as it has been previously observed for other adsorbates [14]. Similar behaviour has been already reported for this type of pristine samples. However, small variation upon adsorption of the Dirac cone, around -360 meV, has been reported for epitaxial graphene on SiC functionalized with nanoparticles. [37] [41].

Action taken: the pinning of the Dirac cone are now better discuss in the text with a new reference. The invariability of the minimum of the Dirac cone is also discussed and compared with other previous works on nanoparticles that shown similar results.

Q4. The geometries of the system along the minimum energy path should be shown along with the corresponding description. This should be done for both nitrogen and oxygen incorporation into the graphene lattice.

We agree with the reviewer that this information is important and has to be added. In the Figure we show the most representative geometries (Initial State, IS, Transition State, TS, and Final State, FS) along the Minimum Energy Path (MEP) obtained within the CI-NEB approach for: (top panel) the dehydrogenation of the OH terminating group of the p-aminophenol molecule and the subsequent linkage of the dehydrogenated p-aminophenol molecule into the graphene lattice via a SAV from the O side; (bottom-left panel) the first dehydrogenation of the NH₂ terminating group of the p-aminophenol molecule; and (bottom-right panel) the second dehydrogenation of the NH₂ terminating group of the p-aminophenol molecule (first dehydrogenation of the remaining NH terminating group) and the subsequent incorporation of the doubly dehydrogenated p-aminophenol molecule into the graphene lattice via a SAV from the N side.

(top panel) In this case the p-aminophenol molecule approaches from the side of the OH terminating group of the molecule. When the molecule is close enough from the surface an O—H stretching phonon is activated (by effect of the temperature and the collision between the surface and molecule electronic clouds) producing the detachment of the H atom from the

OH group, which undergoes to chemisorb “on top” an undersaturated C atom forming the MV. The dehydrogenated p-aminophenol molecule links the remaining terminating O atom to other undersaturated C atom forming the MV. The final state shows the axis of the molecule forming an angle of around 60° with the surface plane. C atoms involved in the adsorption of the detached H atom and the dehydrogenated p-aminophenol molecule show a non-negligible buckling $> 0.3 \text{ \AA}$ off the graphene plane.

(bottom-left panel) In this case the p-aminophenol molecule approaches from the side of the NH_2 terminating group of the molecule. When the molecule is close enough from the surface an N—H stretching phonon in the NH_2 group is activated (by effect of the temperature and the collision between the surface and molecule electronic clouds) producing the detachment of one of the H atoms from the NH_2 group, which undergoes to chemisorb “on top” an undersaturated C atom forming the MV. The final state shows the axis of the molecule forming an angle of around 70° with the surface plane. C atom involved in the adsorption of the detached H atom and the singly dehydrogenated p-aminophenol molecule show a non-negligible buckling $> 0.3 \text{ \AA}$ off the graphene plane.

(bottom-right panel) Once the p-aminophenol molecule has lost its first hydrogen from the NH_2 terminating group, and the detached H atom is chemisorbed “on top” an undersaturated C atom forming the MV, a second dehydrogenation of the remaining NH terminating group is produced: an N—H stretching phonon in the NH remaining group is activated (by effect of the temperature and the collision between the surface and molecule electronic clouds) producing the detachment of the last H atom from the NH group, which again undergoes to chemisorb “on top” other undersaturated C atom forming the MV. In this case the doubly dehydrogenated p-aminophenol molecule incorporates the remaining terminating N atom within the graphene lattice bonded to the three undersaturated C atoms forming the MV in a substitutive manner, in such a way that the N atom fill the MV hole leaving complete the graphene lattice. The final state shows the axis of the molecule forming an angle of around 90° with the surface plane. C atoms involved in the chemisorption of the detached H atoms show a non-negligible buckling $> 0.3 \text{ \AA}$ off the graphene plane. The N atoms, although perfectly incorporated into the graphene lattice, shows a buckling $> 0.6 \text{ \AA}$ off the graphene plane.

Action taken: A sentence has been included in the main text, and the figure has been included as a new section in the supplementary information file (see section 13).

Q5. In Fig.2, the comparison between theory and experiment for different bias voltages is not so convincing. The authors need to explain the figure more carefully in order to convince the readers regarding a good agreement between theoretical and experimental images.

This point is somehow similar to the question Q4.2 from reviewer-1, and we invite the reviewer to read it. We have now explained in a more intuitive way the origin of the small apparent height in the STM images, which could be vague in the previous version. A more complete and accurate description of the origin of the corrugation comes after a comparison of theory with experiments (see fig. 2). We consider that our comparison between the STM experimental and Keldish-Green simulated images for a range of negative bias voltages (-700 , -500 and -300 mV) capture fairly well the fundamental information evinced by the experiment: molecule seen as a bump over the “intact” graphene lattice. It is important to notice that other theoretical approaches to simulate STM images (e.g. Tersoff-Hamann, among others) would not be able to capture the essentials of this particular problem. Typically, other methodologies to simulate STM images take, in a rude first approximation, the total density of

electronic states as proportional to the STM signal without having into account the electronic propagation of the tunnelling current between the tip and the sample. Within our Keldish-Green STM approach we have into account not only the density of states of the sample, but also the density of states of the tip and the “hoppings” between the tip and the sample to propagate the tunnelling current by using the Green-functions formalism. In this particular case, in which we are scanning at negative bias voltages, the fact of just visualizing the graphene morphology in the STM image provides very useful information.

Besides, from the computed density of states profiles we can conclude that the HOMO and the LUMO of the molecule, once it has been incorporated within the graphene lattice, have suffered a significant spreading out due to the chemical interaction with the graphene. The molecular LUMO is remarkably spread out and it extends within a region of around 1 eV width just above the Fermi energy, while the molecular HOMO is also spread out and lies located in a region of around 1 eV width below -1.2 eV from the Fermi level. Our methodology is able to capture this behaviour since the STM images are taken in a range between -1 eV below the Fermi energy and the Fermi level, and the resulting morphology of the STM image is basically formed by the occupied states of the graphene, although slightly altered by the density of states that the molecule induces on it, but not by any electronic states coming purely from the molecule. That is the reason explaining why the theoretical STM images between -700 mV and -300 mV just show electronic states coming from the slightly altered graphene, which is just what the experiments evidence. As an example, a simulated STM image of the system for a bias voltage of +500 mV is the following:

In this case we are tunnelling unoccupied states and capturing part of the spread molecular LUMO, which translates into a visible bump located just in the molecule making up any other signal coming from the graphene.

Taking all that into account, we have compared the calculated corrugation with the experimental one. Although we are aware that these calculations have to be taken cautiously, they indicate that we could expect apparent ranging randomly between 1.7 and 1.9 Å, varying the applied bias from -700 mV to -300 mV, at a currents of about 0.05 nAmps. These values have been obtained by theoretical STM scanning lines along fix trajectories. See fig. S7 in ESI. They have to be compared with the experimentally determined values of 1.6-2 Å for the same range of voltages. The trend is very similar.

Action taken: several actions have been taken to let clear that our agreement is good: New fig. S7 includes calculated values to the experimental points to see that the values are really similar. A paragraph comparing experimental and theoretical corrugation has been included, another explaining quantitatively the origin of the STM images, and finally a better description of the density of states is provided in the main text.

Reviewers' Comments:

Reviewer #1 (Remarks to the Author)

The authors have considered in detail all my comments in their reply and modified in several points the paper and the supporting information.

I think that the paper has improved significantly and that the authors have provided enough evidence to support their conclusions.

In particular I underline that fig. S10 clearly demonstrates the presence of the other group of the molecule.

I suggest to show in the supporting information also the AFM figure presented in response to point Q4.4 in their reply.

Regarding point Q4.3 I think it would have been better to show also an image of the functionalised surface with a bias chosen to show the entire molecule.

I think however that the explanation provided in the text to explain why the entire molecules is not visible is now more convincing (page 13).

There are finally a few typing errors which should be corrected and are listed below.

In conclusion I think that the paper is now acceptable after the authors will have considered the minor points raised in this report.

LIST OF MINOR POINTS:

Fig. 1 caption: SINGLE in place of sibgle.. typical in place of pypical...

Fig.4 caption: if the radius of 1 nm is not TAKEN ...

ref. 24 Celasco, E. et al. . PCCP 18, 18692 (2016) and not Chem Phys. etc.

pag. 19: In our methodology the maximum number of independent covalently linked molecules is about 0.025 %

Check the number please. I guess they mean 0.25 % (i.e. 0.025 ML)

page 10 supporting information: The carrier (hole) density should be of the order of magnitude of some $10^{12}/\text{cm}^2$ to $10^{13}/\text{cm}^2$ (and I guess not $10^{(-12)}/\text{cm}^2$.)

page 16 supporting information: "The results have revealed that the average concentration of p-aminophenol covalently linked to graphene at the vacancy sites is 3.48×10^{-3} (molecules $\times\text{cm}^2$) / C atom."

I think the authors mean 3.48×10^{-3} molecules/C atom.

Fig. S7: top and bottom in place of left and right panels.

Reviewer #2 (Remarks to the Author)

I would like to state that the authors did a good job in the revision of the manuscript and they addressed many issues highlighted by all reviewers. However, I am still against the publication as two key conclusions are not supported satisfactorily. These are related to the mechanism of nitrogen attachment to the graphene lattice and the amount of covalently attached molecules, respectively.

i) Personally, I highly appreciate the effort of the authors to analyze the nature of nitrogen and stability of the bound molecule with XPS. However, the obtained high-resolution N1s spectrum should be analyzed in much more details as the binding energy is considerably lower compared to that expected for graphitic nitrogen. Other possible structural motifs should be excluded (-chemisorbed N₂, -NH₂, pyridinic/pyrrolic nitrogen). I understand the problems with analysis of C1s and O1s spectra due to the specific character of the sample. The newly added S1s spectra after the attachment of aminothiophenol are convincing. In summary, the mechanism based on nitrogen attachment is highly probable but needs more attention (see also point ii)

ii) The total amount of the attached molecules is generally low. Also in this case, I appreciate new experiments (Fig. S9) showing the increasing coverage with increased irradiation time. However, the molecules would be non-covalently attached to the graphene surface. Indeed, the evolved gas analysis (EGA) and TG identifying the molecular species released at higher temperatures (mainly nitrogen bearing functional groups) represent an unambiguous proof for the amount of covalently attached molecules. At the same time, it would bring important findings on the mechanism of the nitrogen action.

I would consider the manuscript for the publication in Nature Communications as soon as the above-mentioned issues are sufficiently addressed.

Reviewer #3 (Remarks to the Author)

The authors have responded to my queries point by point. The response is quite satisfactory. They have modified the paper accordingly. The modified version has more clarity in the text. In my opinion, the work is convincing and is expected to boost up more works along this direction. I recommend publication of the paper in its present form.

Specific answer to reviewers

First of all, I would like to thank again all the reviewers by their critical and careful reading of the manuscript. At the same time we would like to deeply thank them for recognizing the time-consuming effort and dedication by the authors trying to address all their comments, indications and suggestions. With no doubt, they all have strongly helped us to produce a more focused and complete version of the manuscript.

Reviewer 1

In conclusion I think that the paper is now acceptable after the authors will have considered the minor points rose in this report.

We thank the effort and the meticulous analysis performed by this reviewer.

LIST OF MINOR POINTS:

Fig. 1 caption: SINGLE in place of sibgle, typical in place of pypical...

The typos have been fixed.

Fig.4 caption: if the radius of 1 nm is not TAKEN ...

Fixed.

ref. 24 Celasco, E. et al. . PCCP 18, 18692 (2016) and not Chem Phys. etc.

Indeed. We particularly apologize by this remarkable mistake, and thank the referee for realising it. It has been accordingly corrected.

pag. 19: In our methodology the maximum number of independent covalently linked molecules is about 0.025 %. Check the number please. I guess they mean 0.25 % (i.e. 0.025 ML)

Referee is completely right. We have checked and corrected the number.

page 10 supporting information: The carrier (hole) density should be of the order of magnitude of some $10^{12}/\text{cm}^2$ to $10^{13}/\text{cm}^2$ (and I guess not $10^{(-12)}/\text{cm}^2$.)

Thank you very much. The mistake has been corrected.

page 16 sopporting information: "The results have revealed that the average concentration of p-aminophenol covalently linked to graphene at the vacancy sites is 3.48×10^{-3} (molecules $\times\text{cm}^2$) / C atom." I think the authors mean 3.48×10^{-3} molecules/C atom.

Indeed. "cm²" has been removed from the units.

Fig. S7: top and bottom in place of left and right panels.

Fixed.

Reviewer 2

I would like to state that the authors did a good job in the revision of the manuscript and they addressed many issues highlighted by all reviewers. However, I am still against the publication as two key conclusions are not supported satisfactorily. These are related to the mechanism of nitrogen attachment to the graphene lattice and the amount of covalently attached molecules, respectively.

We appreciate the criticism of the reviewer since, also in our opinion, this is the most striking point of the manuscript. However, we cannot escape from the experimental and theoretical facts. In particular, from a wide perspective, we have many different evidences showing specific covalent linking at the vacancy sites.

Nevertheless, stimulated by these comments, we have performed a new set of calculations and new mass spectrometry experiments that we expect can be conclusive and categorical. They are addressed next, in answer at the specific questions.

i) Personally, I highly appreciate the effort of the authors to analyse the nature of nitrogen and stability of the bound molecule with XPS. However, the obtained high-resolution N1s spectrum should be analysed in much more details as the binding energy is considerably lower compared to that expected for graphitic nitrogen. Other possible structural motifs should be excluded (-chemisorbed N2, -NH2, pyridinic/pyrrolic nitrogen). I understand the problems with analysis of C1s and O1s spectra due to the specific character of the sample. The newly added S1s spectra after the attachment of aminothiophenol are convincing. In summary, the mechanism based on nitrogen attachment is highly probable but needs more attention (see also point ii)

The main problem for the core-level XPS assignation of new species is related to the fact that do not exists an absolute value for their binding energies as reference. However, several protocols can be envisaged to try to elucidate the binding energy:

1. First, the most evident by its simplicity are charge arguments
2. On the other hand, looking for existing previous literature to position our binding energy with respect to similar chemical compounds
3. Finally, attempting to perform core-level-shift (CLS) calculations

1. Charge arguments

In our model, although the N is occupying a substitutional place, it remains bonded to the molecule in a sort of sp³ configuration (with a quasi-tetrahedral arrangement involving the 3 C atoms of the SAV within the graphene, the non-negligible buckling expelling out the N atom off the plane, and the C atom of the functionalizing molecule). Moreover, following our computed charge population analysis, N atom accumulates around 0.82 e⁻ and, therefore, in an initial-state approximation, a core-level-shift towards lower binding energies with respect to a N in a graphitic environment can be expected. This effect is what we experimentally observe. Besides, we should not miss the fact that in this quasi-sp³ configuration the N atom is chemically over-coordinated, which could produce a weakening in the bonding and, with it, in the deepest electronic states.

2. Place ourselves using existing literature with the respect to the possible scenarios

First, we have elaborated a table (see below), where most of the related cases have been summarized. We have experimentally checked that we do not have N₂ adsorption, or even physisorbed molecules on the graphene surface (see spectrum in Figure 1g), in which we dosed aminophenol on pristine graphene – without any vacancy – and we were not able to see any N 1s core level peak nor any trace on the STM (see Figure 1g).

Now, after the formation of the vacancies, any remaining species must be covalently bound, as they are able to support high temperature without alteration. We can foresee only 3 possible scenarios:

i) Purely graphitic N. This would be the case if the molecule breaks upon reaction with the vacancy. This case has been deeply studied recently and there exist a bunch of references. We can take as a good reference the one numbered as [39] in the main text ([S42] in Supplementary), where, for sure, they have implanted N within the graphene lattice, and the binding energy for this system would be 400.5 eV. More than 1 eV shifted up with respect to our peak. Statistically, we have found that this value is quite accurate; and using other references we found a dispersion lower than 0.5 eV. Therefore, we can discard that the molecule is broken in our experiment, and that our N is substitutional.

ii) Molecular adsorption by phenol group. In this case the amino group will stand out of the molecule, far from the surface, and it would lead to a configuration that is not very stable by energy arguments, since the oxygen atom is not able to fully saturate electronically the open vacancy (see theoretical mechanism in supplementary section numbered as 13). In this case, we can expect that the binding energy will be, somehow, similar to that obtained on chemisorbed molecules that expose an amino group, as –NH₂ species. As an example, the cysteine molecule, which bound by the S to the gold surface, appears at 401.3 eV, value far away from our experimental observation or the –NH₂ group in biomolecules has been reported at an energy value of 400.1 eV. Therefore, we can discard this possibility.

iii) Amino bonding to the surface due to a molecular cracking. Similarly to the previous case, the reported binding energy for this scenario is 400.1 eV (see table below). This value is far from our experimental XPS observation.

Chemical Species	Graphitic N	R-N-C	-NH ₂	-N=	-NH-	R-NH ₂ Cysteine	Our experiment
References	S42, S43, S44	S45	S46	S46	S46, S47	S48	
Binding Energy (eV)	400.5, 400.6, 400.9	399.7	401.1	398.9	400.1	401.3	399.5

[S42] F. Speck, et al., The quasi-free-standing nature of graphene on H-saturated SiC(0001). Appl. Phys. Lett., 99, 122106 (2011) (*ref. [39] in main text*)

[S43] Lv. Ruitao, et al., Nitrogen-doped graphene: Beyond single substitution and enhanced molecular sensing. Sci. Rep. 2, 1–8 (2012)

[S44] D. Yang, et al., Chemical analysis of graphene oxide films after heat and chemical treatments by X-ray photoelectron and Micro-Raman spectroscopy. Carbon 47, 145–152 (2009)

[S45] J. Choi, et al., Covalent functionalization of epitaxial graphene by azidotrimethylsilane. J. Phys. Chem. C 113, 9433–9435 (2009) (*ref. [40] in main text*)

[S46] E. Mateo-Martí, et al., Self-assembled monolayers of peptide nucleic acids on gold surfaces: A spectroscopic study. Langmuir 21, 9510–9517 (2005)

[S47] A. Garcia-Lekue, Coordinated H-bonding between porphyrins on metal surfaces. *J. Phys. Chem. C* 116, 15378–15384 (2012)

[S48] M. Honda, et al., Electrochemical immobilization of biomolecules on gold surface modified with monolayered L-cysteine. *Thin Solid Films* 556, 307–310 (2014)

Thus, from the previous table, summarizing previous related literature, it is evident that the closest value is the labelled as R-N-C, which can make us to exclude graphitic N, -NH₂ and R-NH₂.

3. First-principles Core-Level-Shift (CLS) calculations

In order to rationalize the origin of the N 1s core-level-shift observed in our XPS experiments we have carried out a set of DFT-based calculations with the plane-wave code QUANTUM ESPRESSO for the following different molecule adsorption configurations:

For the calculations of core level binding energy shifts we have employed the final state approximation, where all the contributions are accounted (initial approximation + difference in screening response between the original and the reference systems). First-principles calculations within this approach have been successfully applied in many different studies ranging from core level shifts in small molecules, nanowires and clusters to surface core level shifts.

Taking graphitic N binding energy to be 400.5 eV (Ref. [39] in the main text and [S42] in Supplementary), the binding energies obtained from our calculations would be (see Supplementary):

Atom	Atom ^{ref}	Binding energy (eV) - TM -	Binding energy (eV) - RRKJ -
N _{4-AP(I)}	N _G	399.27	399.19
N _{4-AP(II)}	N _G	399.83	399.74
N _{4-AP(III)}	N _G	400.75	400.84

According to the binding energy results showed in the previous table the best agreement with our N 1s XPS experiment (399.5 eV) is obtained for the model I (see figure), which correspond to the one we propose in the present study (399.27 and 399.19 eV for TM and RRKJ pseudopotentials, respectively). This value is quite different for other linked forms, as the N₄₋

AP(III) , in which the 4-AP molecule incorporates into the graphene lattice via the oxygen atom (see Figure S14). See further details in Supplementary.

Action taken:

We have included a paragraph on the main text (pages 7 and 8) reinforcing the assignment of our XPS N 1s binding energy with the charge arguments and theoretical calculations. Additionally, we have included a new section in the Supplementary (new section 15), entitled: "Exploring N 1s core level shift by DFT-based calculations" where all the previous information is compiled and explained in full detail. Another section, entitled "N 1s binding energy references" that includes the previous binding energy reference table is given as Supplementary Note 16.

ii) The total amount of the attached molecules is generally low. Also in this case, I appreciate new experiments (Fig. S9) showing the increasing coverage with increased irradiation time. However, the molecules would be non-covalently attached to the graphene surface. Indeed, the evolved gas analysis (EGA) and TG identifying the molecular species released at higher temperatures (mainly nitrogen bearing functional groups) represent an unambiguous proof for the amount of covalently attached molecules. At the same time, it would bring important findings on the mechanism of the nitrogen action.

On the previous version of the manuscript we had given several arguments supporting the covalent bonding towards unspecific adsorption: stable STM images and N 1s core level up to 250°C, binding energy of N 1s peak (experiments, and in the current version theory – see above –), or the use of different amino-contained molecules. Nevertheless, motivated by the concern of the reviewer, we have performed new thermal desorption experiments.

In order to have further probes we have performed mass spectrometry desorption experiments. In our experimental set-up we have the mass-quadrupole head very close to the surface, but, however, some spurious interferences from sample holder and sample manipulator can be expected, as we dose on the whole sample holder. During the experiments, we have submitted the pristine sample to a dose of about 30 Langmuirs of p-AP

at RT. After the dose we have applied a temperature ramp recording the amount of different masses that have desorbed. The result is presented in the previous new figure, also included in the new section 14 of the Supplementary.

In the previous figure (included in Supplementary as Figure S13) we show the recorded TPD spectra of p-AP desorbing from a surface of SLG epitaxially grown on SiC(0001) before and after creating SAVs. Also shown are the difference spectra from TPD signals before and after SAV creation coming from p-AP (with a mass of 109 amu) and two of its fragments (with 92 and 93 amu). Two features labelled as A and B can be observed in the TPD spectra. Feature A is the most intense and best defined of the two. Due to: i) its higher intensity, ii) its recognisable line shape of a first-order desorption process, and iii) an absence of significant increase of the total pressure in the vacuum chamber, we can assign blue-shaded zone A in the Figure to p-AP molecules originating from a homogeneous source close to the mass spectrometer ionization source i.e., the sample surface. The difference spectra show virtually no signal in the range of feature A, so we conclude that the p-AP molecules that are not bonded to a SAV desorb in the same way, independently of the presence of the SAV. The peak temperature of 60°C coincides well with STM experiments, where it was observed that a small annealing up to 100°C – or waiting for several hours at RT – improved significantly the stability of the measurements after depositing p-AP.

On the other hand, feature B (red-shaded zone in the Figure) exhibits a very broad ill-defined and less intense structure, and can be attributed to background signal stemming from molecules reaching the spectrometer indirectly and from inhomogeneous sources i.e., originating from sample holder and sample manipulator. This observation also coincides with a noticeable increase of the total pressure in the vacuum chamber due to molecules desorbing from the various surfaces. This background signal is expected due to the aforementioned absence of a Feulner cup. A small difference can be observed in the difference spectra within the range of zone B. As the sample has been moved in between the two TPD runs, some small changes in the background are expected, framed within the uncertainties of our TPD setup.

This information has been included in the current version further developed and explained in detail in the new section 14 of the Supplementary.

Action taken:

We have included a new paragraph in the main text (pag. 8) specifically mentioning the absence of unspecific adsorption and refereeing to the Thermal desorption experiments at the supplementary file. We have included a new section in the Supplementary explaining the details and the results of this new experiment (section 14, entitled: “Thermal Programmed Desorption”), as well as the previous Figure with its corresponding caption.

As a direct consequence of this new experiment, the person who carried it out appears now in the revised version of the manuscript as a co-author: Dr. Koen Lauwaet.

I would consider the manuscript for the publication in Nature Communications as soon as the above-mentioned issues are sufficiently addressed.

We expect that these new discussion, experiments and theoretical calculations would have dissipated the doubts of the reviewer, and this version could be accepted for publication in its current form.

Reviewers' Comments:

Reviewer #2 (Remarks to the Author)

The authors addressed all my concerns very carefully. Thus, I recommend the manuscript for publication in Nature Communications. I have just one minor request: very recent publications describing highly efficient functionalization starting from fluorographene towards graphene covalently modified with thio-, cyano-, or carboxy- groups should be cited.